# SGN: Shifted Window-Based Hierarchical Variable Grouping for Multivariate Time Series Classification

**Zenan Ying[1],    Jinke Wang[1],    Zhi Zheng[1]*,    Tong Xu[1],**
**Wei Chen[1],    Qi Liu[1],    Huijun Hou[2]**

[1]State Key Laboratory of Cognitive Intelligence, University of Science and Technology of China
[2]Nio
`{znying, wjk1008, chenweicw, liuqilq}@mail.ustc.edu.cn`
`{tongxu, zhengzhi97}@ustc.edu.cn`
`huijun.hou@nio.com`

## Abstract

Multivariate time series (MTS) classification has attracted increasing attention across various domains. Existing methods either decompose MTS into separate univariate series, ignoring inter-variable dependencies, or jointly model all variables, which may lead to over-smoothing and loss of semantic structure. These limitations become particularly pronounced when dealing with complex and heterogeneous variable types. To address these challenges, we propose **SwinGroupNet** (SGN), which explores a novel perspective for constructing variable interaction and temporal dependency. Specifically, SGN processes multi-scale time series using (1) *Variable Group Embedding* (VGE), which partitions variables into groups and performs independent group-wise embedding; (2) *Multi-Scale Group Window Mixing* (MGWM), which reconstructs variable interactions by modeling both intra-group and inter-group dependencies while extracting multi-scale temporal features; and (3) *Periodic Window Shifting and Merging* (PWSM), which exploits inherent periodic patterns to enable hierarchical temporal interaction and feature aggregation. Extensive experiments on diverse benchmark datasets from multiple domains demonstrate that SGN consistently achieves state-of-the-art performance, with an average improvement of 4.2% over existing methods. We release the source code at `https://github.com/colison/SGN`.

## 1 Introduction

Multivariate time series (MTS) consist of multiple temporal variables, each representing distinct dynamic patterns over time. MTS classification, which aims to analyze and model these temporal signals jointly to extract meaningful patterns for decision-making, has demonstrated significant importance across a wide range of application domains, including meteorology [1, 2], healthcare [3, 4, 5], industrial monitoring [6, 7], and human activity recognition [8, 9]. In recent years, many models have been developed specifically for temporal data analysis [10, 11, 12, 13, 14, 15, 16], achieving impressive performance across diverse applications. Among these, Convolutional Neural Networks (CNNs) have demonstrated continual development in the time series domain [17, 18, 19, 20] due to their strong ability to extract local features along the temporal dimension and have been shown to strike an effective balance between performance and computational efficiency [21].

In multivariate time series processing, in addition to modeling along the temporal dimension, capturing the dependencies among variables is equally critical. However, the complex and intertwined

---

*Corresponding author.

39th Conference on Neural Information Processing Systems (NeurIPS 2025).

dependencies among variables pose a significant challenge for effective modeling. Existing works have mainly adopted two strategies for modeling variable relationships, namely independent modeling and mixed modeling. Specifically, as shown in Figure 1(a), independent modeling-based methods [22, 23] treat multivariate data as a collection of univariate series and process each variable separately, thereby neglecting the inter-variable interdependencies. In contrast, as shown in Figure 1(b), mixed modeling-based methods [24, 25] jointly model all variables by mixing them together. However, when dealing with complex and heterogeneous variable types, this strategy blurs the semantic distinctions between variables [26], leading to excessive smoothing and making it difficult for the model to capture meaningful relationships [27].

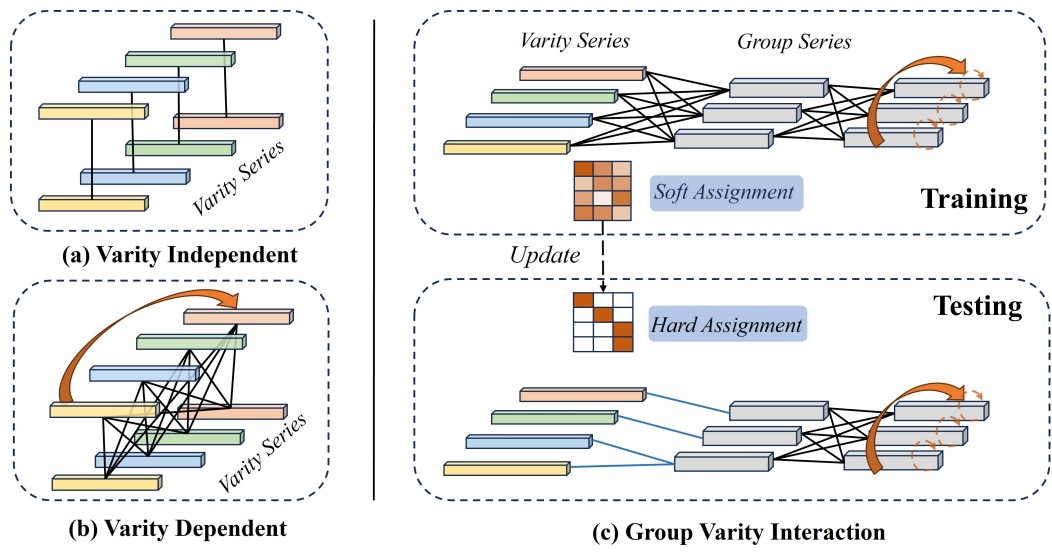

Figure 1: Illustration for Variety Interaction. (a) Variety Independent. (b) Variety Dependent. (c) Our Group Variety Interaction.

To address these challenges, in this paper, we introduce a novel perspective for modeling variable dependencies by transforming global variable interactions into structured intra-group and inter-group relationships. Specifically, we design the *Variable Group Embedding* (VGE) module, which partitions variables into groups based on an assignment matrix derived from their intrinsic similarity and performs independent group-wise embedding. As illustrated in Figure 1(c), the module employs soft assignments during training to allow flexible learning, and switches to hard assignments during inference to ensure stable group structures. This design is particularly effective when dealing with heterogeneous multivariate time series, as it enhances the model's capacity to capture diverse variable characteristics and their interactions.

Moreover, we segment the input sequence into small periodic windows and propose the *Multi-Scale Group Window Mixing* (MGWM) module, which reconstructs variable interactions by modeling both intra-group and inter-group dependencies while extracting temporal features at multiple scales. To further enhance the modeling of temporal dynamics, we introduce the *Periodic Window Shifting and Merging* (PWSM) module, which leverages inherent periodic patterns to enable hierarchical temporal interaction and feature aggregation. By integrating both variable and temporal perspectives, we present the **SwinGroupNet (SGN)** architecture, which performs structured variable grouping and interaction modeling in the variable dimension, and efficient multi-scale feature extraction in the temporal dimension. Meanwhile, SGN effectively balances performance and computational efficiency while capturing long-range dependencies and complex temporal patterns in multivariate time series. Extensive experiments on diverse datasets from various domains demonstrate that SGN consistently achieves state-of-the-art results. Our main contributions are summarized as follows:

- We propose a novel variable interaction strategy that transforms variable relationships into intra-group and inter-group dependencies based on variable grouping, unveiling underlying patterns and enhancing the interpretability of variable relationships.

- Extensive experiments demonstrate that SGN consistently outperforms existing state-of-the-art methods, with an average accuracy gain of 4.2% and achieves near-perfect performance on several benchmarks, with accuracy approaching 100%.

- We exploit the inherent periodic characteristics of time series to enable hierarchical temporal feature extraction, allowing CNN-based models to capture long-term dependencies while maintaining high efficiency.

## 2 Related Work

### 2.1 Convolution in Time Series Analysis

In recent years, convolutional methods have seen increasing adoption in time series analysis. CNN-based models typically focus on local patterns, with convolutional kernels adept at capturing localized features from the input. To expand the receptive field and capture long-range temporal dependencies, TCNs [28, 29] employ dilated convolutions. MICN [17] integrates both local and global features through multi-scale extraction, enabling the modeling of complex temporal patterns. TimesNet [18] transforms univariate time series into a two-dimensional format via periodic decomposition and applies 2D convolutions to capture intra-period and inter-period patterns. ModernTCN [19] further extends the receptive field by utilizing large convolutional kernels. TVNet [20] reshapes 1D sequences into 3D representations to extract hierarchical information across temporal dimensions. Despite these advancements, existing convolutional models often overlook the intricate dependencies between local and global contexts as well as heterogeneous variable interactions. Therefore, there remains considerable room for improvement in developing models that can comprehensively capture the complexity of multivariate time series.

### 2.2 Variable Modeling Strategies in Multivariate Time Series

In multivariate time series analysis, variable modeling strategies aim to capture the dependencies among multiple variables. Given the inherently complex inter-variable relationships in such data, explicitly modeling these dependencies is crucial for learning comprehensive representations and enhancing model performance. Existing approaches can be broadly categorized into two strategies. Methods such as PatchTST [22], RLinear [16] treat each variable independently, enabling the model to specialize per variable. In contrast, models like iTransformer [24], Crossformer [25] mix all variables to learn cross-variable correlations, while CrossGNN [30] leverages graph structures to enhance the representation and understanding of variable interactions. These approaches have collectively advanced the field of multivariate time series analysis. Prior studies [31, 32, 33, 34] suggest that variable-independent models often offer greater model capacity, whereas variable-mixing strategies tend to exhibit improved robustness. However, striking a balance between these strategies remains a challenge. Recent work [35, 36, 37, 38]has explored the utility of variable clustering, yet in the time series domain, it remains unclear how best to translate variable-to-cluster relationships into meaningful interactions and how clustering can be leveraged to enhance inter-variable modeling. This remains an open and promising direction for further research.

## 3 Method

As shown in Figure 2, we propose **SwinGroupNet (SGN)**, a model for multivariate time series classification that captures structured interactions among variable groups. Specifically, we design a *Variable Group Embedding* (VGE) module that segments the input sequence into short periodic windows and embeds variables into groups based on their similarity. Next, the *Multi-Scale Group Window Mixing* (MGWM) module extracts features along both the temporal and variable dimensions. Furthermore, the *Periodic Window Shifting and Merging* (PWSM) module exploits inherent periodic patterns to enable hierarchical interaction and aggregation of temporal features. Finally, a projection layer produces the classification output. The following sections provide detailed descriptions of each component.

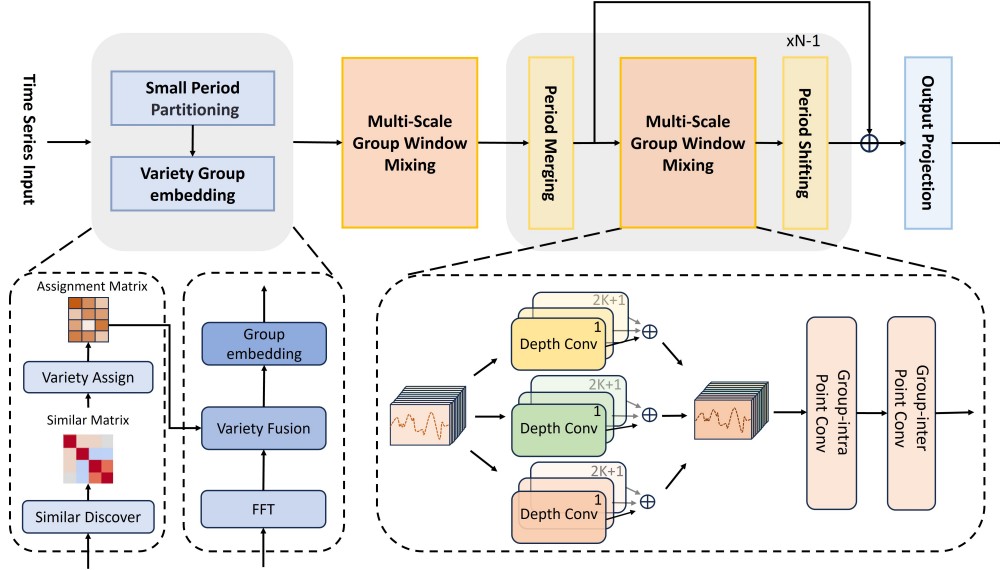

Figure 2: An illustration of the SwinGroupNet architecture.

## 3.1 Variable Group Embedding

In this module, we first group the different variables of each sample and perform interactions within and across groups to capture the relationships among variables. For heterogeneous multivariate data, directly mixing all variables, including those with large differences, can cause over-smoothing and confuse the semantic meanings between variables. On the other hand, modeling each variable independently would neglect the dependencies among them. By clustering variables into groups, our method strikes a balance between these two extremes: it preserves the distinct properties of each variable while simultaneously mitigating interference among variables.

Specifically, for the input time series $X \in \mathbb{R}^{C \times L}$, where $L$ represents the time length and $C$ denotes the number of variables, the feature channels $C$ are dynamically assigned to predefined groups $G$ using the Gumbel-Softmax allocation mechanism [39, 40]. Given a categorical distribution, to avoid the collapse problem, instead of using randomly initialized unnormalized class logits, we compute the Brownian Distance Covariance (BDC) [41] between all variables. The specific formula is as follows:

$$\mathrm{dCov}(X_1, X_2) = \sqrt{\frac{1}{n^2} \sum_{i,j=1}^{n} A'_{ij} B'_{ij}}. \tag{1}$$

Here, $A'_{ij}$ and $B'_{ij}$ denote the elements of the centered distance matrices for $X_1$ and $X_2$, respectively, which are derived from the Euclidean distances between observations in $X_1$ and $X_2$. Given the number of groups $G$, the similarity is then computed based on the BDC distance, and K-means clustering is applied to obtain $G$ class logits vectors $\Pi = [\pi_1, \pi_2, \ldots, \pi_G]$, where each vector $\pi_i \in \mathbb{R}^C$. For more details, refer to Appendix A. And the Gumbel-Softmax sampling is defined as:

$$M_{ji} = \frac{\exp\left((\log \pi_{ij} + \epsilon_{ij})/\tau\right)}{\sum_{k=1}^{G} \exp\left((\log \pi_{kj} + \epsilon_{kj})/\tau\right)}, \quad j = 1, 2, \ldots, C, \quad i = 1, 2, \ldots, G. \tag{2}$$

Here, $\epsilon \sim \mathrm{Gumbel}(0, 1)$ denotes noise sampled from a standard Gumbel distribution, used to enhance the exploration during assignment. The parameter $\tau > 0$ is the temperature that controls the smoothness of the resulting distribution. It is gradually annealed following an exponential decay schedule during training.

The input tensor $X$ is grouped according to the assignment matrix $M \in \mathbb{R}^{C \times G}$, where $M_{ji}$ denotes the soft assignment probability of variable $j$ to group $i$. We use Softmax-based soft assignment

during training and one-hot hard assignment during evaluation. To avoid inconsistent assignments and unstable optimization, we introduce a variable similarity regularization term to guide the Gumbel-based assignment towards more reasonable groupings. During the training process, the cosine similarity $S_{ij}$ is dynamically computed with Variety $X_i$ and $X_j$ after normalizing the variables to obtain the similarity matrix $S \in \mathbb{R}^{C \times C}$, while incorporating the assignment matrix into the loss function. The specific regularization loss formula is as follows:

$$\mathcal{L}_{\text{sim}} = \sum_{i,j} S_{ij} \cdot \|M_i - M_j\|^2. \tag{3}$$

Therefore, the final loss is given by $\mathcal{L} = \mathcal{L}_{\text{task}} + \beta \mathcal{L}_{\text{sim}}$ and $\beta$ is a regularization parameter for balancing classification accuracy and cluster quality. Meanwhile, to facilitate effective intra-group and inter-group interactions after variable grouping, we perform independent embeddings for each variable group.

## 3.2 Multi-Scale Group Window Mixing

**Small Periodic Window Partitioning**. Specifically, periodic patterns in time series often rely on information in the frequency domain. To capture this, we first apply the Fast Fourier Transform (FFT) to the input data $X$ and compute the mean of the amplitude spectrum [18].

$$A = \text{Avg}(\text{Amp}(\text{FFT}(X))). \tag{4}$$

To achieve a better trade-off between performance and efficiency, instead of selecting the top-$K$ periods with the highest amplitudes, we choose the smallest period among the top-$K$ candidates as the cycle window.

$$P = \min \left\{ \frac{L}{f_i} \ \middle| \ f_i \in \text{argTopK}(A), \ i = 1, 2, \ldots, K \right\}. \tag{5}$$

Each $\frac{L}{f_i}$ corresponds to the $i$-th dominant frequency component. Given the selected period window $P$, we divide the time series into $N = \lceil \frac{L}{P} \rceil$ segments, where zero-padding is applied at the end if $L$ is not divisible by $P$, obtain the output sequence as $X \in \mathbb{R}^{C \times N \times P}$. Considering the extensibility of periodic patterns, we further incorporate multiple scales by merging additional period windows corresponding to the remaining Top-$K$ dominant frequencies. This approach allows us to extract multi-scale periodic information and enhance computational efficiency.

**Multi-Scale Group Window Extracting**. Given a multivariate time series $X \in \mathbb{R}^{C \times N \times P}$, to better extract temporal and variable features, We follow the design proposed by Liu [42, 43], adopting a combination of depthwise convolution and pointwise convolution to separate temporal and variable information. Along the temporal dimension, we apply multiple convolution kernels of different receptive fields within each period window to perform multi-scale feature extraction like [18, 44, 45]. The extracted features from different scales are then aggregated through average pooling to enhance the robustness of the output representations. The detailed formulation is given as follows:

$$Y = \frac{1}{K} \sum_{i=0}^{K-1} \mathcal{C}_g^{(k_i)}(X), \quad k_i = 2i + 1, \quad \forall i \in \{0, 1, \ldots, K-1\}, \tag{6}$$

where $\mathcal{C}_g^{(k_i)}(\cdot)$ denotes a grouped convolution operation applied at a specific scale parameterized by the kernel size $k_i$. Across the variable dimension, we decouple the full variable interactions into intra-group and inter-group interactions, effectively mitigating the oversmoothing issue commonly observed in deep architectures. Specifically, we apply separate pointwise convolutions within each group and across different groups, respectively, to jointly learn fine-grained dependencies among variables within a group and higher-order relationships among different groups. The detailed structure of the Multi Swin Window block is illustrated in Figure 2.

### 3.3 Periodic Window Shifting and Merging

**Periodic Window Shifting**. Although the above procedure successfully extracts features from small periodic windows, it lacks connections across different windows, which limits interactions across periodic segments. Inspired by the Swin Transformer [46], we leverage the phase shift property of periodic signals to enable efficient cross-window communication. Specifically, we first perform a cyclic left shift of the input time series $X$ by $P/2$ units, where $P$ is the periodic window length. We then redivide the shifted sequence into windows and extract the features accordingly. The process can be formulated as follows:

$$X_{\text{shifted}} = \mathcal{T}_{-\frac{P}{2}}\left(\text{Reshape}(X, (C, N \cdot P))\right), \tag{7}$$

where $\mathcal{T}_{-\frac{P}{2}}$ denotes a cyclic left shift operation by $P/2$ units. After feature extraction, we apply a cyclic right shift to restore the original temporal order. This operation introduces cross-window interactions while maintaining computational efficiency. By alternating between standard window partitioning and shifted partitioning, the model captures both intra-window and inter-window dependencies within the same periodic scale. Specifically, as shown in Figure 3, this approach effectively enhances the model's ability to capture complex temporal patterns.

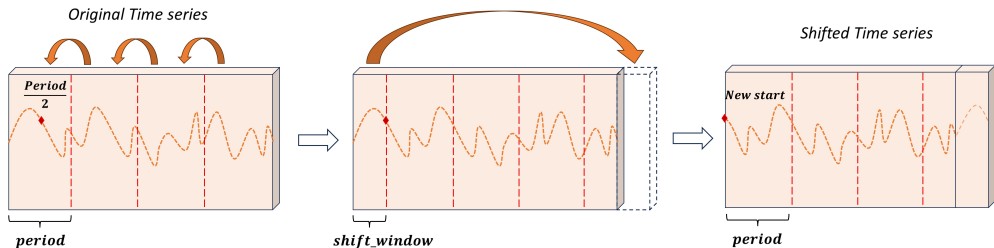

Figure 3: Illustration of Periodic Window Shifting Block.

**Periodic Window Merging.** Based on the extensibility of periodic patterns, we further merge adjacent periodic windows to capture features over extended cycles. Specifically, two neighboring periodic segments are combined to generate representations corresponding to $2P$-length cycles. Notably, such extended periods often correspond to dominant frequencies among the top-$k$ spectral components. To enhance computational efficiency, we employ an adaptive merging strategy based on the number of detected periodic windows. When the number of periods exceeds four, we adopt an exponentially decaying merging scheme, in which non-overlapping adjacent windows are progressively merged. In contrast, when fewer than four periods are present, we switch to a linearly decaying merging strategy that allows overlapping windows, thereby enabling more effective feature extraction from larger periodic structures.

## 4 Experiments

**Dataset.** To evaluate the effectiveness of the proposed SwinGroupNet model, we conduct extensive experiments on a diverse set of multivariate time series datasets. The detailed main dataset information is provided in Table 1. Furthermore, to comprehensively assess the generalization ability of our model, we additionally select 10 multivariate datasets from the UEA Time Series Classification Archive, which was introduced by Bagnall [47]. For additional details on data characteristics and preprocessing, please refer to Appendix B.

**Baseline.** We include 10 state-of-the-art time series methods as baselines to comprehensively evaluate the performance of our proposed method. These methods include six Transformer-based models: Reformer [48], iTransformer [24], Crossformer [25], FEDformer [14], PatchTST [22], Medformer [49]; One MLP-based model: DLinear [15]; And three CNN-based models: MICN [17], TimesNet [18], ModernTCN [19].

**Implementation.** In the main datasets, we adopt six evaluation metrics: Accuracy, Precision, Recall, F1 Score, AUROC, and AUPRC. The training process is conducted using five different random seeds

Table 1: Summary of the four benchmark datasets used in our experiments. The table lists the number of samples, classes, channels, length, sampling rate and modality.

| Datasets | Samples | Timestamps | Channel | Class | Sampling Rate | Modality | FileSize |
|---|---|---|---|---|---|---|---|
| TDBRAIN | 6,240 | 256 | 33 | 2 | 256 Hz | EEG | 571MB |
| PTB-XL | 191,400 | 250 | 12 | 5 | 250 Hz | ECG | 4.28GB |
| FLAAP | 13,123 | 100 | 6 | 10 | 100 Hz | HAR | 60MB |
| UCI-HAR | 10,299 | 128 | 9 | 6 | 50 Hz | HAR | 91MB |

(41–45) on fixed training, validation, and test splits to compute the mean and standard deviation of the results. For the 10 multivariate datasets from the UEA Time Series Classification Archive, we follow the standard preprocessing protocol established by Wu [18]. See Appendix C.1 for more implementation details.

Table 2: Performance comparison on Main Datasets from different domains. **Bold** values represent the best performance, and underlined values indicate the second-best scores.

| Dataset | Model | Accuracy | Precision | Recall | F1 Score | AUROC | AUPRC |
|---|---|---|---|---|---|---|---|
| TDBRAIN | Reformer | $87.92 \pm 2.01$ | $88.64 \pm 1.40$ | $87.92 \pm 2.01$ | $87.85 \pm 2.08$ | $96.30 \pm 0.54$ | $96.40 \pm 0.45$ |
| | Crossformer | $81.56 \pm 2.19$ | $81.97 \pm 2.25$ | $81.56 \pm 2.19$ | $81.50 \pm 2.20$ | $91.20 \pm 1.78$ | $91.51 \pm 1.71$ |
| | FEDformer | $78.13 \pm 1.98$ | $78.52 \pm 1.91$ | $78.13 \pm 1.98$ | $78.04 \pm 2.01$ | $86.56 \pm 1.86$ | $86.48 \pm 1.99$ |
| | iTransformer | $74.67 \pm 1.06$ | $74.71 \pm 1.06$ | $74.67 \pm 1.06$ | $74.65 \pm 1.06$ | $83.37 \pm 1.14$ | $83.73 \pm 1.27$ |
| | PatchTST | $79.25 \pm 3.79$ | $79.60 \pm 4.09$ | $79.25 \pm 3.79$ | $79.20 \pm 3.77$ | $87.95 \pm 4.96$ | $86.36 \pm 6.67$ |
| | Medformer | $89.62 \pm 0.81$ | $89.68 \pm 0.78$ | $89.62 \pm 0.81$ | $89.62 \pm 0.81$ | $96.41 \pm 0.35$ | $96.51 \pm 0.33$ |
| | Dlinear | $54.73 \pm 2.14$ | $54.79 \pm 2.48$ | $54.73 \pm 2.14$ | $54.62 \pm 2.13$ | $55.83 \pm 2.36$ | $54.73 \pm 1.98$ |
| | Timesnet | $95.08 \pm 0.56$ | $95.11 \pm 0.58$ | $95.08 \pm 0.56$ | $95.08 \pm 0.56$ | $98.92 \pm 0.19$ | $98.95 \pm 0.19$ |
| | MICN | $90.92 \pm 2.24$ | $91.37 \pm 1.82$ | $90.92 \pm 2.24$ | $90.89 \pm 2.46$ | $97.58 \pm 1.84$ | $97.62 \pm 1.93$ |
| | ModernTCN | $87.60 \pm 2.03$ | $88.10 \pm 1.38$ | $87.60 \pm 2.03$ | $87.54 \pm 2.13$ | $95.72 \pm 0.87$ | $95.87 \pm 0.94$ |
| | **SGN (Ours)** | **$99.90 \pm 0.10$** | **$99.89 \pm 0.11$** | **$99.90 \pm 0.10$** | **$99.90 \pm 0.10$** | **$100.00 \pm 0.00$** | **$100.00 \pm 0.00$** |
| PTB-XL | Reformer | $71.72 \pm 0.43$ | $63.12 \pm 1.02$ | $59.20 \pm 0.75$ | $60.69 \pm 0.18$ | $88.80 \pm 0.24$ | $64.72 \pm 0.47$ |
| | Crossformer | $73.30 \pm 0.14$ | $65.06 \pm 0.35$ | $61.23 \pm 0.33$ | $62.59 \pm 0.14$ | $90.02 \pm 0.06$ | $67.43 \pm 0.22$ |
| | FEDformer | $57.20 \pm 9.47$ | $52.38 \pm 6.09$ | $49.04 \pm 7.26$ | $47.89 \pm 8.44$ | $82.13 \pm 4.17$ | $52.31 \pm 7.03$ |
| | iTransformer | $69.28 \pm 0.22$ | $59.59 \pm 0.45$ | $54.62 \pm 0.18$ | $56.20 \pm 0.19$ | $86.71 \pm 0.10$ | $60.27 \pm 0.21$ |
| | PatchTST | $73.23 \pm 0.25$ | $65.70 \pm 0.64$ | $60.82 \pm 0.76$ | $62.61 \pm 0.34$ | $89.74 \pm 0.19$ | $67.32 \pm 0.22$ |
| | Medformer | $72.87 \pm 0.23$ | $64.14 \pm 0.42$ | $60.60 \pm 0.46$ | $62.02 \pm 0.37$ | $89.66 \pm 0.13$ | $66.39 \pm 0.22$ |
| | Dlinear | $45.49 \pm 0.03$ | $20.25 \pm 9.92$ | $20.10 \pm 0.09$ | $12.78 \pm 0.26$ | $50.63 \pm 0.12$ | $20.75 \pm 0.13$ |
| | Timesnet | $71.80 \pm 0.53$ | $62.73 \pm 0.88$ | $59.53 \pm 0.99$ | $60.72 \pm 0.49$ | $88.27 \pm 0.65$ | $63.53 \pm 1.07$ |
| | MICN | $67.33 \pm 0.36$ | $56.98 \pm 0.93$ | $51.90 \pm 0.81$ | $53.29 \pm 0.52$ | $85.63 \pm 0.29$ | $57.70 \pm 0.49$ |
| | ModernTCN | $72.85 \pm 0.19$ | $63.68 \pm 0.43$ | $60.20 \pm 0.82$ | $61.33 \pm 0.64$ | $89.54 \pm 0.31$ | $66.00 \pm 0.46$ |
| | **SGN (Ours)** | **$73.80 \pm 0.34$** | **$65.88 \pm 0.47$** | **$62.17 \pm 0.61$** | **$63.43 \pm 0.49$** | **$90.25 \pm 0.17$** | **$67.76 \pm 0.58$** |
| FLAAP | Reformer | $70.88 \pm 0.88$ | $71.47 \pm 0.77$ | $70.22 \pm 1.13$ | $70.19 \pm 1.02$ | $95.27 \pm 0.28$ | $74.64 \pm 1.24$ |
| | Crossformer | $76.33 \pm 0.81$ | $76.25 \pm 0.93$ | $76.15 \pm 0.84$ | $76.14 \pm 0.88$ | $96.93 \pm 0.13$ | $80.25 \pm 0.69$ |
| | FEDformer | $68.30 \pm 2.06$ | $69.18 \pm 0.96$ | $67.60 \pm 2.12$ | $66.80 \pm 2.96$ | $94.15 \pm 0.76$ | $70.85 \pm 3.12$ |
| | iTransformer | $75.83 \pm 0.49$ | $75.70 \pm 0.65$ | $75.82 \pm 0.56$ | $75.57 \pm 0.53$ | $96.70 \pm 0.14$ | $80.32 \pm 0.64$ |
| | PatchTST | $56.23 \pm 0.28$ | $56.21 \pm 0.69$ | $55.45 \pm 0.24$ | $55.57 \pm 0.35$ | $88.92 \pm 0.09$ | $58.40 \pm 0.28$ |
| | Medformer | $74.00 \pm 2.37$ | $74.53 \pm 2.48$ | $73.84 \pm 2.61$ | $73.57 \pm 2.55$ | $96.58 \pm 0.60$ | $78.91 \pm 2.90$ |
| | Dlinear | $30.26 \pm 1.46$ | $27.46 \pm 1.06$ | $28.20 \pm 0.99$ | $25.71 \pm 0.49$ | $70.76 \pm 0.49$ | $26.70 \pm 0.23$ |
| | Timesnet | $73.79 \pm 0.94$ | $73.55 \pm 0.78$ | $73.57 \pm 0.92$ | $72.82 \pm 0.96$ | $95.70 \pm 0.12$ | $77.30 \pm 0.84$ |
| | MICN | $52.63 \pm 0.59$ | $51.74 \pm 0.76$ | $51.45 \pm 0.50$ | $50.84 \pm 0.56$ | $88.35 \pm 0.32$ | $48.81 \pm 0.45$ |
| | ModernTCN | $71.66 \pm 1.69$ | $72.23 \pm 1.45$ | $71.55 \pm 1.66$ | $71.37 \pm 1.46$ | $95.04 \pm 0.30$ | $73.47 \pm 1.99$ |
| | **SGN (Ours)** | **$80.81 \pm 0.41$** | **$80.68 \pm 0.32$** | **$80.35 \pm 0.50$** | **$80.35 \pm 0.41$** | **$97.42 \pm 0.10$** | **$85.73 \pm 0.37$** |
| UCI-HAR | Reformer | $90.00 \pm 0.63$ | $90.10 \pm 0.71$ | $90.14 \pm 0.75$ | $89.92 \pm 0.63$ | $98.97 \pm 0.08$ | $95.86 \pm 0.29$ |
| | Crossformer | $90.66 \pm 1.02$ | $90.83 \pm 0.98$ | $90.69 \pm 1.02$ | $90.68 \pm 1.04$ | $99.14 \pm 0.15$ | $96.08 \pm 0.61$ |
| | FEDformer | $86.90 \pm 3.46$ | $88.57 \pm 0.99$ | $87.58 \pm 3.15$ | $87.71 \pm 2.24$ | $97.66 \pm 1.08$ | $92.87 \pm 1.64$ |
| | iTransformer | $93.47 \pm 0.15$ | $93.59 \pm 0.23$ | $93.49 \pm 0.14$ | $93.46 \pm 0.16$ | $99.53 \pm 0.02$ | $97.91 \pm 0.01$ |
| | PatchTST | $86.83 \pm 0.68$ | $87.59 \pm 0.68$ | $87.15 \pm 0.78$ | $87.17 \pm 0.77$ | $98.43 \pm 0.12$ | $93.57 \pm 0.50$ |
| | Medformer | $90.17 \pm 0.52$ | $90.36 \pm 0.66$ | $90.30 \pm 0.68$ | $90.27 \pm 0.66$ | $99.10 \pm 0.07$ | $95.84 \pm 0.75$ |
| | Dlinear | $61.28 \pm 1.26$ | $60.89 \pm 1.34$ | $59.37 \pm 1.36$ | $58.83 \pm 1.00$ | $84.70 \pm 0.35$ | $59.29 \pm 0.41$ |
| | Timesnet | $92.71 \pm 0.50$ | $92.78 \pm 0.46$ | $92.86 \pm 0.53$ | $92.75 \pm 0.51$ | $99.25 \pm 0.04$ | $96.87 \pm 0.24$ |
| | MICN | $86.23 \pm 0.73$ | $86.72 \pm 0.66$ | $86.26 \pm 0.73$ | $86.22 \pm 0.73$ | $98.63 \pm 0.15$ | $92.94 \pm 1.19$ |
| | ModernTCN | $92.75 \pm 2.03$ | $92.96 \pm 2.31$ | $92.88 \pm 2.16$ | $92.80 \pm 2.18$ | $99.35 \pm 0.19$ | $96.98 \pm 0.91$ |
| | **SGN (Ours)** | **$95.62 \pm 0.72$** | **$95.79 \pm 0.69$** | **$95.61 \pm 0.72$** | **$95.64 \pm 0.70$** | **$99.77 \pm 0.08$** | **$98.96 \pm 0.39$** |

## 4.1 Result of Main Datasets

**Setups.** In our main experimental setup, the training, validation, and testing sets are partitioned either by subject or according to a fixed ratio, depending on the dataset characteristics. Samples from each subject are assigned to the respective sets following a fixed allocation ratio. Importantly, samples

from the same subject are restricted to a single subset to avoid any data leakage. This design ensures the independence and objectivity of model training and evaluation.

**Results.** As shown in Table 2, SGN consistently outperforms ten strong baseline models across four benchmark datasets of different types, achieving the best performance on all evaluation metrics. On average, in terms of accuracy, our method surpasses the second-best approach **4.2%**. Specifically, SGN yields significant improvements on TDBRAIN, FLAPP, and UCI-HAR, outperforming the second-best models by **4.8%**, **4.5%**, and **2.2%**, highlighting the advantage of modeling variable dependencies through variable grouping. Notably, on the TDBRAIN dataset, SGN achieves an impressive approaching **100%** accuracy. However, the performance gain on the PTB-XL dataset is relatively marginal. This can be attributed to the inherent variable similarity structure: TDBRAIN exhibits clear boundaries among variable clusters, enabling effective group-based modeling, while PTB-XL presents high similarity across variables with blurred boundaries, causing most variables to be grouped together. As a result, group interactions degenerate into a conventional mixed-variable approach. The variable similarity visualizations for each dataset are provided in Appendix B.6.

| Dataset / Model | W.+MUSE | M.-FCN | TapNet | ShapeNet | TodyNet | SVPT | ShapeFormer | MPTSNet | SGN (Ours) |
|---|---|---|---|---|---|---|---|---|---|
| ArticularyWordRecognition | **99.0** | 97.3 | 98.7 | 98.7 | 98.7 | 99.3 | 99.0 | 97.7 | **99.0** |
| AtrialFibrillation | 33.3 | 26.7 | 33.3 | 40.0 | 46.7 | 40.0 | 53.3 | 53.3 | **66.7** |
| BasicMotions | **100.0** | 95.0 | **100.0** | **100.0** | **100.0** | **100.0** | **100.0** | **100.0** | **100.0** |
| Cricket | **100.0** | 91.7 | 95.8 | 98.6 | **100.0** | **100.0** | 94.4 | 94.4 | **100.0** |
| DuckDuckGeese | 57.5 | 67.5 | 57.5 | **72.5** | 58.0 | 70.0 | 64.0 | 68.0 | 64.0 |
| Epilepsy | **100.0** | 76.1 | 97.1 | 98.7 | 97.1 | 98.6 | 98.6 | 97.1 | 97.8 |
| EthanolConcentration | 13.3 | 37.3 | 32.3 | 31.2 | 35.0 | 33.1 | 41.1 | 43.3 | **44.5** |
| ERing | 43.0 | 13.3 | 13.3 | 13.3 | 91.5 | 93.7 | 87.4 | 94.4 | **95.9** |
| FaceDetection | 54.5 | 54.5 | 55.6 | 60.2 | 62.7 | 51.2 | 65.8 | 69.8 | **70.3** |
| FingerMovements | 49.0 | 58.0 | 53.0 | 58.9 | **67.6** | 60.0 | 55.0 | 64.0 | 64.0 |
| HandMovementDirection | 36.5 | 36.5 | 37.8 | 33.8 | 64.9 | 39.2 | 41.9 | 63.5 | **75.7** |
| Handwriting | **60.5** | 28.6 | 35.7 | 45.1 | 43.6 | 43.3 | 30.2 | 34.4 | 50.4 |
| Heartbeat | 72.7 | 66.3 | 75.1 | 75.6 | 75.6 | 79.0 | **81.5** | 75.6 | 77.1 |
| Libras | 87.8 | 85.6 | 85.0 | 85.6 | 85.0 | 88.3 | **95.5** | 87.2 | 83.9 |
| LSST | 59.0 | 37.3 | 56.8 | 59.0 | 61.5 | **66.6** | 63.8 | 60.4 | 63.7 |
| MotorImagery | 50.0 | 51.0 | 59.0 | 61.0 | 64.0 | **65.0** | N/A | **65.0** | **65.0** |
| NATOPS | 87.0 | 88.9 | 93.9 | 88.3 | 97.2 | 90.6 | 96.1 | 94.4 | **98.3** |
| PenDigits | 94.8 | 97.8 | 98.0 | 97.7 | 98.7 | 98.3 | **99.1** | 98.9 | **99.1** |
| PEMS-SF | N/A | 69.9 | 75.1 | 75.1 | 78.0 | 86.7 | N/A | **94.2** | 88.4 |
| PhonemeSpectra | 19.0 | 11.0 | 17.5 | 29.8 | **30.9** | 17.6 | 29.3 | 14.4 | 23.1 |
| RacketSports | **93.4** | 80.3 | 86.8 | 88.2 | 80.3 | 84.2 | 88.8 | 87.5 | **93.4** |
| SelfRegulationSCP1 | 71.0 | 87.4 | 65.2 | 78.2 | 89.8 | 88.4 | 91.8 | 92.8 | **93.9** |
| SelfRegulationSCP2 | 46.0 | 47.2 | 55.0 | 57.8 | 55.0 | 60.0 | 56.1 | 57.2 | **60.6** |
| StandWalkJump | 33.3 | 6.7 | 40.0 | 53.3 | 46.7 | 46.7 | **66.7** | 53.3 | 53.3 |
| UWaveGestureLibrary | 91.6 | 89.1 | 89.4 | 90.6 | 85.0 | **94.1** | 90.0 | 88.1 | 92.2 |
| **Average Rank** | 6.04 | 7.60 | 6.68 | 5.14 | 4.64 | 3.94 | 3.70 | 4.20 | **2.56** |
| **Number of Top-1** | 6 | 0 | 1 | 2 | 4 | 5 | 5 | 3 | **14** |
| **Wins** | 17 | 23 | 23 | 19 | 20 | 15 | 13 | 15 | - |
| **Draws** | 4 | 0 | 1 | 2 | 3 | 3 | 4 | 4 | - |
| **Loses** | 3 | 2 | 1 | 4 | 2 | 7 | 6 | 3 | - |

Table 3: Performance comparison with the recent advanced MTSC-dedicated models on 25 UEA datasets. In the table, 'N/A' indicates that the results for the corresponding method could not be obtained due to memory or computational limitations

## 4.2 Result of UEA Multivariate Datasets

**Setups.** We selected multivariate time series datasets from the UEA Multivariate Time Series Classification Archive. All datasets have been preprocessed and standardized using established preprocessing techniques. To ensure a comprehensive comparison, we incorporated several state-of-the-art baseline methods into our evaluation framework like LSTNet [50], LightTS [51], Rocket [52], LSSL [53], Flowformer [54], MTSMixer [55], TVNet [20], TimeMixer++ [56] on the UEA-10 datasets. Meanwhile, we incorporated WEASEL+MUSE [57], MLSTM-FCN [58], TapNet [59], ShapeNet [60], TodyNet [61], SVPT [62], ShapeFormer [63] and MPTSNet [64] on the UEA-25 datasets, under the same experimental settings.

**Results of UEA-10.** As illustrated in Figure 4, we present the average classification accuracy across ten UEA datasets. It can be observed that SGN surpasses the current best-performing model, TimeMixer++, and consistently outperforms other categories of models. Notably, MLP-based models

exhibit subpar performance on classification tasks, which can be attributed to their lack of explicit modeling of dependencies among variables. In contrast, CNN-based models perform better due to their strong local feature extraction capabilities and ability to capture variable interactions. Complete results are provided in Appendix C.3.

**Results of UEA-25.** Our proposed SGN model consistently demonstrates superior performance across the 25 UEA multivariate time series classification datasets compared with eight state-of-the-art baseline models, as shown in Table 3. It achieves the best average rank of 2.56, the highest number of first-place results (14 datasets), and an overall best win–loss record against all competitors. Notably, SGN outperforms all other models in more than half of the datasets, while maintaining competitive performance in the remaining ones. Furthermore, it exhibits strong generalization ability across diverse domains, including human activity recognition, healthcare, and speech recognition tasks.

## 4.3 Ablation Studies

**Ablation on Variable Grouping.** To evaluate the effectiveness of our proposed variable grouping strategy, we conduct ablation studies on the main datasets by removing specific components (w/o). The results, as shown in Table 4, demonstrate that performing both intra-group and inter-group interactions outperforms the strategy of directly mixing all variables. This confirms the necessity and effectiveness of our structured variable interaction design. Furthermore, we observe that inter-group interaction yields better performance than intra-group interaction alone, indicating that the model benefits more from capturing diverse dependencies across groups than from modeling redundant information among similar variables.

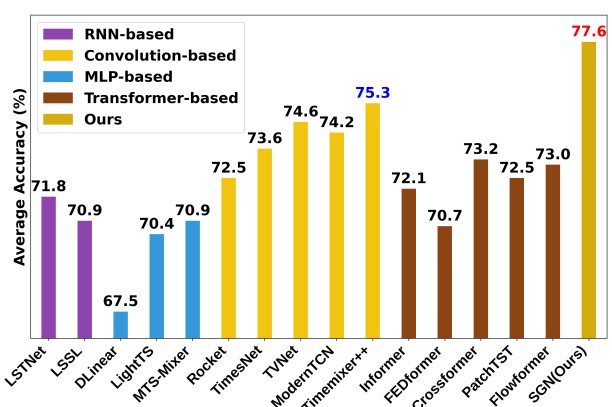

Figure 4: UEA Classification Result of UEA-10 datasets.

Table 4: Ablation Study on Variable Grouping Module across Main Datasets.

| Dataset | TDBrain | | PTB-XL | | FLAAP | | UCI-HAR | |
|---|---|---|---|---|---|---|---|---|
| Metric | Accuracy | F1 | Accuracy | F1 | Accuracy | F1 | Accuracy | F1 |
| **Variable Group Embedding & Group Interaction** | **99.90** | **99.90** | **73.80** | **63.43** | **80.81** | **80.35** | **95.62** | **95.64** |
| w/o Variable Grouping Embedding | 92.16 | 91.64 | 73.05 | 62.46 | 76.85 | 76.36 | 93.58 | 93.58 |
| w/o Group-intra Interaction | 99.81 | 99.81 | 72.19 | 61.02 | 73.82 | 73.63 | 91.62 | 91.69 |
| w/o Group-inter Interaction | 99.83 | 99.83 | 71.70 | 60.66 | 70.60 | 70.21 | 88.20 | 88.15 |

**Ablation on Periodic Window Shifting and Merging.** This module is designed to validate the effectiveness of extracting features using small periodic windows by leveraging periodic properties. We conduct experiments on the main datasets used in the previous sections, and the detailed results are presented in Table 5. As observed, the impact of periodic fusion is particularly significant, confirming its ability to effectively capture global patterns through multi-period aggregation. Additional ablation studies on hyperparameter sensitivity and efficiency analysis are provided in Appendix D and Appendix E, respectively.

Table 5: Ablation Study on Periodic Window Module across Main Datasets.

| Dataset | TDBrain | | PTB-XL | | FLAAP | | UCI-HAR | |
|---|---|---|---|---|---|---|---|---|
| Metric | Accuracy | F1 | Accuracy | F1 | Accuracy | F1 | Accuracy | F1 |
| **Periodic Window Shifting & Merging** | **99.90** | **99.90** | **73.80** | **63.43** | **80.81** | **80.35** | **95.62** | **95.64** |
| w/o Periodic Window Shifting | 99.79 | 99.79 | 73.32 | 62.50 | 78.05 | 77.66 | 87.05 | 87.22 |
| w/o Periodic Window Merging | 99.58 | 99.58 | 72.81 | 62.31 | 69.68 | 68.77 | 77.79 | 77.83 |

### 4.4 Model Analysis.

**Analysis of Variable Clustering.** As illustrated in Figure 5(a), the initial similarity matrix of variables in the TDBRAIN dataset computed using the BDC distance reveals distinct block structures. These structures indicate inherent similarity among subsets of variables even before model training. Figure 5(b) presents the Pearson correlation matrix computed from the model outputs under the best-performing setting without variable grouping. The results reveal highly diverse and complex relationships among variables, indicating that dependencies exist but are not explicitly structured.

In contrast, Figure 5(c) shows the transformed correlation structure under the variable grouping strategy, where correlations are aggregated at both intra-group and inter-group levels. After training, the intra-group correlations are significantly stronger, suggesting tight dependency within groups, while inter-group correlations are considerably weaker, indicating minimal redundancy across groups. These results demonstrate that variety grouping not only alleviates complex inter-variable dependencies and potential redundancies, but also enhances interpretability and performance in modeling multivariate relationships.

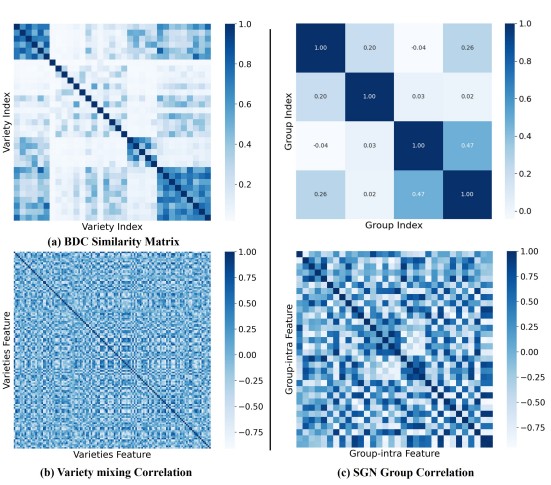

(a) BDC Similarity Matrix

(b) Variety mixing Correlation

(c) SGN Group Correlation

Figure 5: Visualization of TDBRAIN Variety mixing and Variety Grouping matrices.

**Analysis of Periodic Shifting and Merging.** Figure 6 shows the temporal correlation across SGN layers on the UCI-HAR dataset using learned periodic windows. As observed, in the first layer, the temporal dependencies are primarily concentrated around local neighboring time points.

In contrast, in the last layer, each time point exhibits high correlation with almost all points within the periodic window, demonstrating that SGN effectively integrates hierarchical periodic information. This enables the model to extend the local feature extraction strength of convolutional networks to a global temporal context, enhancing its capacity for comprehensive temporal understanding. Additional visualization results of other main datasets and experimental analyses are provided in Appendix F for further reference.

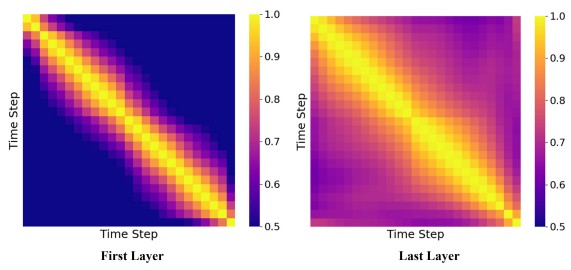

First Layer

Last Layer

Figure 6: Visualization of time correlation under UCI-HAR dataset.

## 5 Conclusion

In this paper, we propose SwinGroupNet, a novel and effective framework for multivariate time series (MTS) classification. By leveraging the Variable Group Embedding strategy, we convert variable-level interactions into structured group-based representations. The Multi-Scale Group Window Mixing mechanism further enhances interaction modeling by capturing both intra-group and inter-group dependencies, while simultaneously extracting multi-scale temporal features to enrich temporal representations. Furthermore, the Periodic Window Shifting and Merging approach integrates hierarchical periodic information, enabling the model to better capture dynamic temporal patterns. SGN achieves state-of-the-art performance across diverse datasets spanning multiple domains. Limitations and potential directions for future research are discussed in Appendix G.

## Acknowledgments and Disclosure of Funding

This work was supported in part by the grants from National Natural Science Foundation of China (No.62222213, U22B2059), in part by the Postdoctoral Fellowship Program and China Postdoctoral Science Foundation under Grant Number BX20250387. This work was also supported by USTC-NIO Smart Electric Vehicle Joint Lab.

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

# A  Variety Assign Logits

Previous studies commonly adopt the *Pearson correlation coefficient* to measure dependencies between variables, which is effective for capturing linear relationships. The Pearson correlation between two variables $\mathbf{x}, \mathbf{y} \in \mathbb{R}^n$ is defined as:

$$\rho(\mathbf{x}, \mathbf{y}) = \frac{\sum_{i=1}^{n}(x_i - \bar{x})(y_i - \bar{y})}{\sqrt{\sum_{i=1}^{n}(x_i - \bar{x})^2} \cdot \sqrt{\sum_{i=1}^{n}(y_i - \bar{y})^2}}, \tag{8}$$

where $\bar{x}$ and $\bar{y}$ denote the sample means of $\mathbf{x}$ and $\mathbf{y}$, respectively. While this approach performs well under linear assumptions, it fails to capture complex nonlinear dependencies—an issue particularly prominent in multivariate time series, where variable interactions can be intricate and highly nonlinear.

To address this limitation, we adopt the *Brownian Distance Covariance (BDC)* as an alternative. Unlike Pearson correlation, BDC is a nonparametric statistical measure that can detect both linear and nonlinear dependencies without assuming any specific distribution. It provides a more general and powerful tool for modeling complex interactions in multivariate time series.The empirical BDC between $\mathbf{x}$ and $\mathbf{y}$ is defined as:

$$\mathcal{R}_n(\mathbf{x}, \mathbf{y}) = \frac{\mathcal{V}_n(\mathbf{x}, \mathbf{y})}{\sqrt{\mathcal{V}_n(\mathbf{x}, \mathbf{x}) \cdot \mathcal{V}_n(\mathbf{y}, \mathbf{y})}}. \tag{9}$$

Here, $\mathcal{V}_n(\mathbf{x}, \mathbf{y})$ quantifies the strength of dependence between $\mathbf{x}$ and $\mathbf{y}$, with a value of zero implying statistical independence. This quantity is computed based on the following formulation:

$$\mathcal{V}_n^2(\mathbf{x}, \mathbf{y}) = \frac{1}{n^2} \sum_{i=1}^{n} \sum_{j=1}^{n} A_{ij} B_{ij}, \tag{10}$$

where $A_{ij}$ and $B_{ij}$ are double-centered distance matrices derived from $\mathbf{x}$ and $\mathbf{y}$, respectively. Then double-centered using the following formula:

$$A_{ij} = \|x_i - x_j\| - \frac{1}{n} \sum_{j'=1}^{n} \|x_i - x_{j'}\| - \frac{1}{n} \sum_{i'=1}^{n} \|x_{i'} - x_j\| + \frac{1}{n^2} \sum_{i'=1}^{n} \sum_{j'=1}^{n} \|x_{i'} - x_{j'}\|, \tag{11}$$

where $\| \cdot \|$ denotes the Euclidean norm. The same procedure applies to matrix $B$ to obtain the centered version $B_{ij}$. Finally, the normalized BDC value lies within the interval $[0, 1]$, where 1 indicates perfect dependence and 0 indicates independence. Importantly, the BDC measure is strictly zero if and only if the variables are independent in the population. Therefore, it serves as a robust and comprehensive dependency metric, particularly valuable for analyzing complex, nonlinear relationships in multivariate time series.

After computing the Brownian Distance Covariance (BDC) matrix, we apply the K-means clustering algorithm on the rows of the BDC matrix to construct a *pre-assignment matrix*. Each row corresponds to a variable's dependency pattern across all other variables. Let $\mathcal{R} \in [0, 1]^{C \times C}$ denote the BDC correlation matrix among $C$ variables. We treat each row $\mathcal{R}_{i:} \in \mathbb{R}^C$ as a feature vector representing variable $i$, and perform K-means clustering [65] to partition the $d$ variables into $G$ clusters:

$$\pi = \text{KMeans}(\mathcal{R}, G), \tag{12}$$

where $\pi \in \{1, \ldots, G\}^C$ is the cluster assignment vector. Based on these assignments, we construct a pre-assignment matrix $P \in \{0, 1\}^{C \times G}$, where each row indicates the cluster membership of the corresponding variable using one-hot encoding. This pre-assignment matrix serves as an initial grouping prior for subsequent model components, such as channel interaction modules or hierarchical aggregation mechanisms.

# B    Datasets

## B.1    TDBRAIN Dataset

The TDBrain dataset, referenced in [66], is a large-scale EEG time series dataset containing recordings from 1,274 subjects using 33 channels. Each subject participated in two trials: one with eyes open and another with eyes closed. The dataset includes a total of 60 diagnostic labels, allowing for multi-label classification as each subject may be associated with multiple conditions. In this paper, we utilize a subset of the dataset comprising 25 subjects diagnosed with Parkinson's disease and 25 healthy controls, all under the eyes-closed condition. Each eyes-closed trial is segmented into non-overlapping 1-second windows, each containing 256 time points. Segments shorter than 1 second are discarded. This preprocessing results in a total of 6,240 samples. Each sample is tagged with a subject ID to indicate its origin.

## B.2    PTB-XL Dataset

The PTB-XL dataset [67] is a large-scale public ECG time series dataset collected from 18,869 subjects, each with 12-channel recordings and annotated with one or more of five labels, including four heart disease categories and one healthy control. Since each subject may have multiple trials, we remove subjects whose diagnoses vary across trials to ensure label consistency, resulting in 17,596 subjects retained. Each trial is a 10-second ECG segment, available in both $100\,Hz$ and $500\,Hz$ sampling rates. In our study, we use the $500\,Hz$ version, downsampled to $250\,Hz$ and normalized using a standard scaler. We then segment each trial into non-overlapping 1-second samples (250 time steps per sample), discarding any segment shorter than 1 second. This preprocessing yields a total of 191,400 samples. For model training, we adopt a subject-independent split, allocating 60%, 20% and 20% of subjects (and their corresponding samples) to the training, validation, and test sets, respectively.

## B.3    FLAAP Dataset

The FLAAP dataset [68] is a human activity recognition (HAR) dataset collected using smartphone-based inertial sensors, specifically accelerometers and gyroscopes placed at the waist of subjects. It records ten distinct daily activities performed by eight subjects, with data continuously captured between February 1st and May 31st, 2022, at a sampling rate of $100\,Hz$. Unlike many existing HAR datasets that focus primarily on activity classification, FLAAP emphasizes discovering associated patterns within activities, aiming to better reflect the structure of Activities of Daily Living (ADL). Each activity is segmented into fixed-length windows, producing a total of 13,123 samples with 6 sensor channels and 100 time steps per sample. In our experiments, the dataset is divided into 60% for training, 20% for validation, and 20% for testing. The dataset serves as a valuable benchmark for studying representation learning, pre-processing effects, domain transfer, and activity association mining in multivariate time series.

## B.4    UCI-HAR Dataset

The UCI-HAR dataset [69] is a widely used benchmark for human activity recognition (HAR), collected using smartphone-based inertial sensors. It contains recordings from 30 subjects performing six different daily activities (walking, walking upstairs, walking downstairs, sitting, standing, and lying) while carrying a smartphone equipped with a tri-axial accelerometer and gyroscope. Data were collected at a sampling rate of $50\,Hz$ and then preprocessed by applying noise filters and segmenting the continuous signal into fixed-length windows of 2.56 seconds (128 time steps) with a 50% overlap. Each segment is labeled with the corresponding activity. In our study, we use the processed version with 9 selected channels and 128 timestamps per sample, resulting in a total of 10,299 labeled samples. The dataset is split into training and test sets based on a certain ratio.

## B.5    UEA Classification Dataset

The UEA dataset [47] is a comprehensive collection of multivariate time series samples spanning a wide range of application domains, primarily designed for classification tasks. It includes diverse recognition scenarios such as facial, gesture, and action recognition, as well as audio classification.

Beyond these, it serves practical purposes in areas like industrial monitoring, health surveillance, and medical diagnostics, with particular attention to cardiac data analysis. Typically, the dataset is structured into 10 distinct classes. Table 6 provides detailed classification statistics, highlighting the dataset's versatility and broad applicability across multiple domains.

Table 6: Datasets and mapping details of UEA dataset.

| Dataset | Sample Numbers (train, test) | Variable Number | Series Length |
|---|---|---|---|
| EthanolConcentration | (261, 263) | 3 | 1751 |
| FaceDetection | (5890, 3524) | 144 | 62 |
| Handwriting | (150, 850) | 3 | 152 |
| Heartbeat | (204, 205) | 61 | 405 |
| JapaneseVowels | (270, 370) | 12 | 29 |
| PEMS - SF | (267, 173) | 963 | 144 |
| SelfRegulationSCP1 | (268, 293) | 6 | 896 |
| SelfRegulationSCP2 | (200, 180) | 7 | 1152 |
| SpokenArabicDigits | (6599, 2199) | 13 | 93 |
| UWaveGestureLibrary | (120, 320) | 3 | 315 |

## B.6 Datasets BDC Similarity Visualization

Figure 7 displays the similarity matrices computed using Brownian Distance Covariance (BDC) across different datasets. The x-axis and y-axis represent the variable (channel) indices.

Figure 7: BDC Similarity Visualization of Main and UEA Datasets.

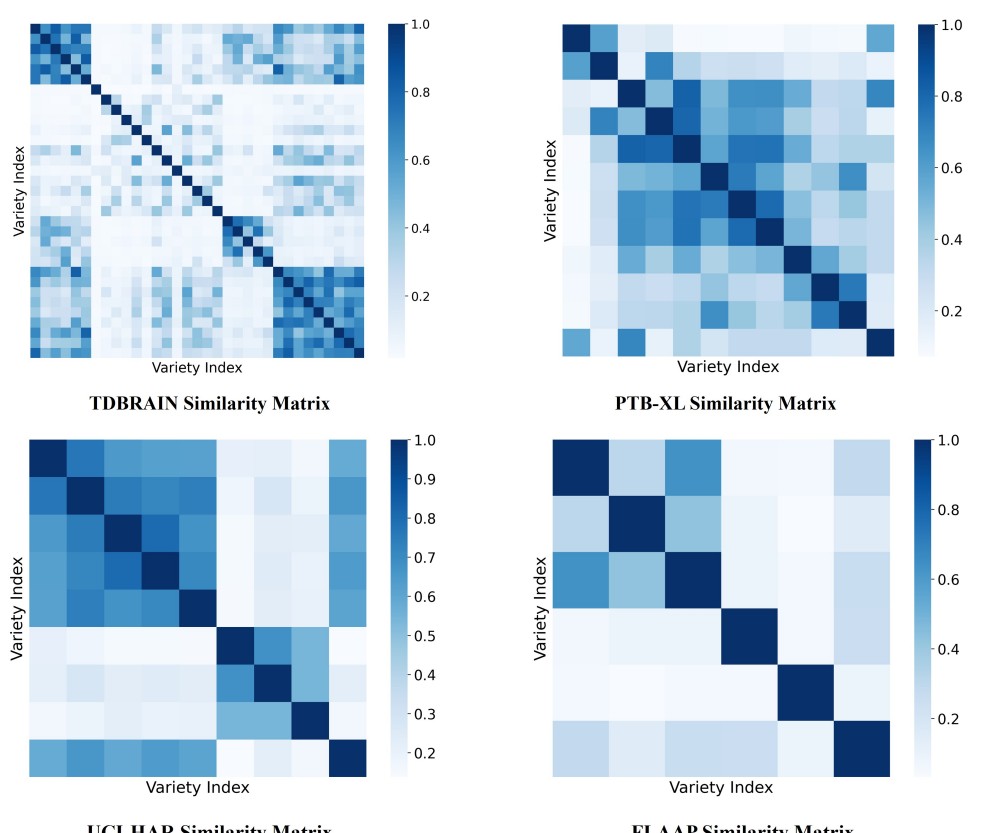

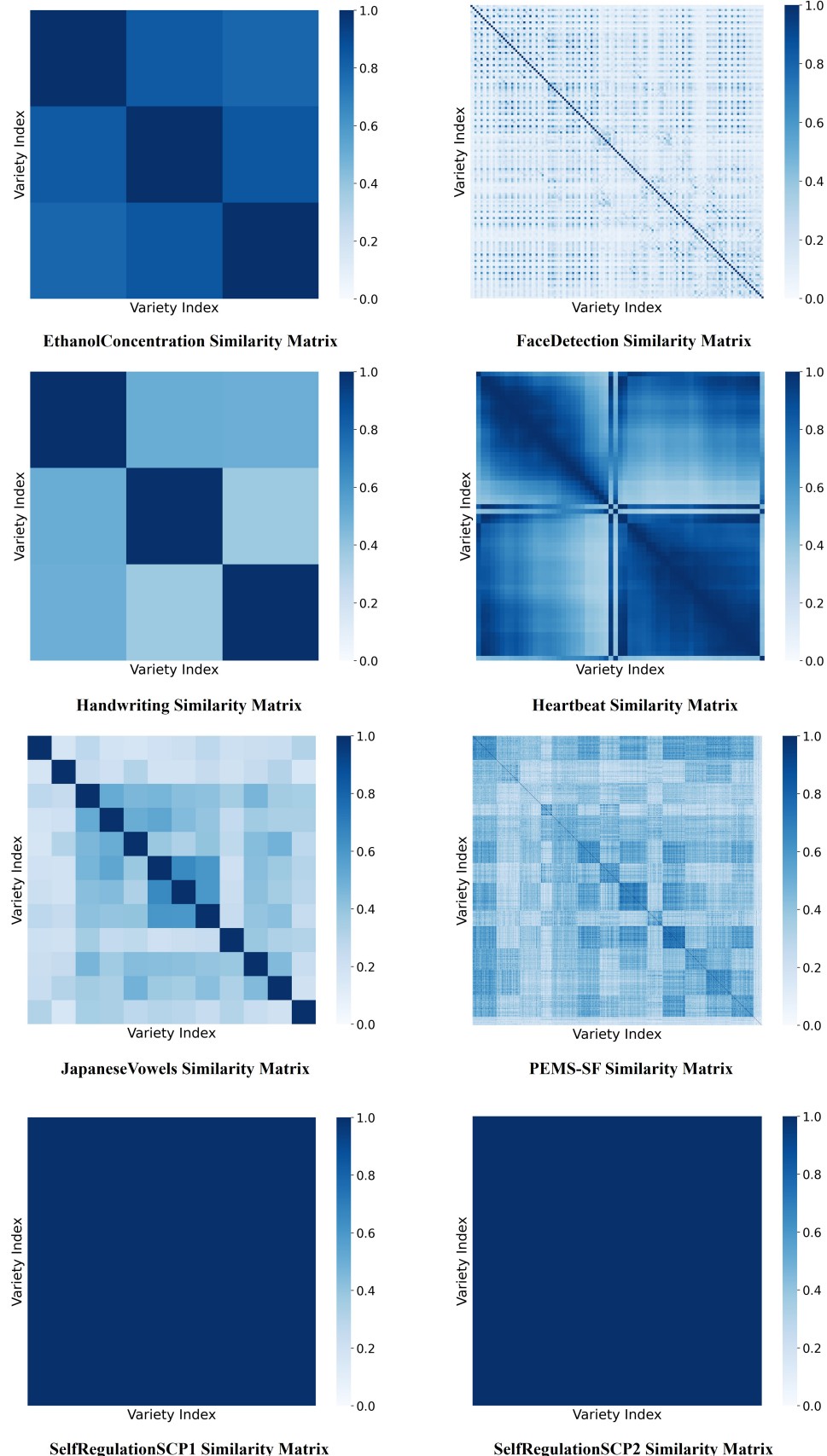

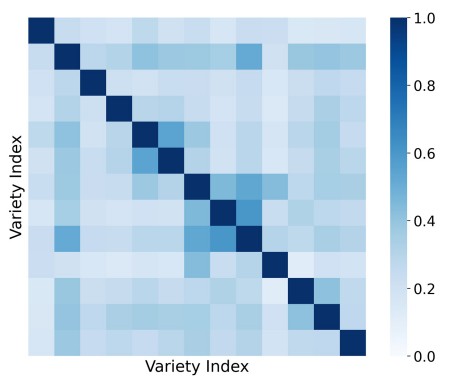
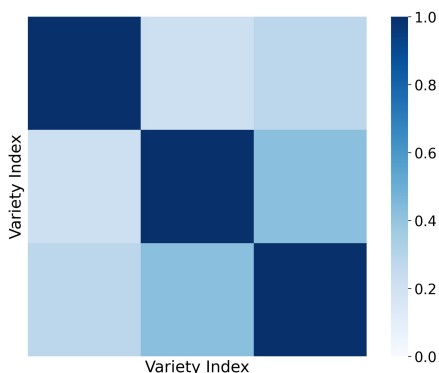

**SpokenArabicDigits Similarity Matrix**          **UWaveGestureLibrary Similarity Matrix**

# C  Experiments Details

## C.1  Main Datasets Implement Details

**Implementation Details.** All methods are implemented within a unified framework to ensure fair comparison. Specifically, we re-implement each approach using a consistent training strategy. We compare our model against a variety of state-of-the-art time series models, including six Transformer-based models, one MLP-based model, and three CNN-based models as baselines. For baseline implementations, we adopt the official code and follow the recommended best configurations. The learning rate is fixed at 0.0001, and the Adam [70] optimizer is employed for all experiments. The batch sizes are set according to the dataset: {32, 256, 32, 32} for TDBrain, PTB-XL, UCI-HAR, and FLAAP, respectively. In the data preprocessing stage, we adopt the processing pipeline from Wang[49] to ensure consistency and comparability across datasets. All models are trained for 100 epochs using five different random seeds (41 to 45), and we report the average results along with the standard deviations. To prevent overfitting, we adopt an early stopping strategy based on the F1 score on the validation set. All experiments are conducted using four NVIDIA RTX 4090 GPUs (24GB memory) with the PyTorch framework [71].

**SwinGroupNet (Our Method).** We perform clustering on multivariate time series based on the similarity matrix derived from BDC, generating an initial assignment matrix (see Appendix A for details). Different similarity thresholds result in different numbers of groups. We then apply Gumbel-Softmax sampling to obtain a differentiable assignment matrix, followed by independent embedding for each group. In our proposed *Multi-Swin Periodic Window*, multi-scale feature extraction is conducted on the input, and information from different layers is fused to produce the final output. The learning rate is set to 0.001, and an early stopping strategy is adopted. The experimental configurations on the main datasets, including the number of variable groups, the regularization parameter $\beta$, embedding dimension, model depth, number of kernels in depthwise convolution, period window size, and channel expansion ratio, are summarized in Table 7.

Table 7: Experiment configuration of SGN.

| Dataset | # Groups | #$\beta$ | # Embedding Dim | # Layers | # Kernels | # Period Window | #Channel Ratio |
|---|---|---|---|---|---|---|---|
| TDBRAIN | 4 | 0.1 | 32 | 5 | 7 | 26 | 2 |
| PTB-XL | 4 | 0.1 | 64 | 5 | 7 | 25 | 2 |
| UCI-HAR | 6 | 0.1 | 64 | 4 | 7 | 32 | 2 |
| FLAAP | 5 | 0.1 | 64 | 2 | 7 | 50 | 2 |

## C.2  Baselines Details

To evaluate the effectiveness of our proposed method, we selected a set of strong baseline models that cover a wide range of architectural paradigms. Specifically, for the main benchmark datasets, we include convolution-based models such as MICN [17], ModernTCN [19], and TimesNet [18]; Transformer-based models including iTransformer [24], Reformer [48], FedFormer [14], Crossformer

[25], PatchTST [22], and MedFormer [49]; and MLP-based models like DLinear [15]. These models have demonstrated strong capabilities in temporal modeling and provide a solid foundation for comparative analysis. To further validate our method, we also conducted experiments on the UEA datasets by including additional baselines such as CNN-based models (TVNet [20], TimeMixer++ [56], Rocket [52]), RNN-based models (LSTNet [50], LSSL [53]), and MLP-based models (LightTS [50], MTS-Mixer [55]).

## C.3 Full result of UEA Datasets

**Implementation Details.** Our method is trained using the cross-entropy loss, with classification accuracy (%) adopted as the evaluation metric. The model is initialized with a learning rate of $10^{-2}$, and an early stopping strategy is applied to prevent overfitting. The symbol "$*$" in Transformer-based models indicates the specific model name (e.g., *former may refer to Informer, Crossformer, etc.). Detailed results are presented in Table 8. As shown in the table, MLP-based models exhibit relatively poor performance. In contrast, CNN-based models demonstrate superior results by effectively extracting local features and incorporating various strategies to capture global patterns.

Table 8: Performance comparison of various models on different datasets with accuracy metrics for classification.

| Datasets / Models | RNN-based | | Convolution-based | | | | | MLP-based | | | Transformer-based | | | | | SGN |
|---|---|---|---|---|---|---|---|---|---|---|---|---|---|---|---|---|
| | LSTNet | LSSL | Rocket | TimesNet | ModernTCN | TVnet | Timemixer++ | DLinear | LightTS | MTS-Mixer | In. | PatchTST | Cross. | Flow. | FED. | (Ours) |
| | 2018 | 2022 | 2020 | 2023 | 2023 | 2025 | 2025 | 2024 | 2022 | 2023 | 2023 | 2022 | 2023 | 2022 | 2022 | |
| EthanolConcentration | 39.9 | 31.1 | 45.2 | 35.7 | 36.3 | 35.6 | 39.9 | 36.2 | 29.7 | 33.8 | 31.6 | 32.8 | 38.0 | 33.8 | 31.2 | 44.5 |
| FaceDetection | 65.7 | 66.7 | 64.7 | 66.0 | 68.0 | 71.2 | 71.8 | 68.0 | 67.5 | 70.2 | 67.0 | 68.7 | 66.1 | 67.6 | 66.0 | 70.3 |
| Handwriting | 25.8 | 24.6 | 58.8 | 32.1 | 27.0 | 32.7 | 26.5 | 26.1 | 27.1 | 33.8 | 32.8 | 29.8 | 30.1 | 33.8 | 28.0 | 50.4 |
| Heartbeat | 77.1 | 72.2 | 75.6 | 75.4 | 77.1 | 78.1 | 79.1 | 76.1 | 73.6 | 76.6 | 80.5 | 76.2 | 77.6 | 77.6 | 73.7 | 77.1 |
| JapaneseVowels | 98.1 | 98.4 | 96.2 | 98.4 | 98.2 | 98.9 | 97.9 | 96.2 | 96.2 | 94.3 | 98.9 | 97.9 | 99.1 | 98.9 | 98.4 | 98.9 |
| PEMS-SF | 86.7 | 86.1 | 75.1 | 89.6 | 89.0 | 91.1 | 91.0 | 88.7 | 87.3 | 90.1 | 81.5 | 89.2 | 90.2 | 86.0 | 80.9 | 88.4 |
| SelfRegulationSCP1 | 84.0 | 90.8 | 90.8 | 91.8 | 91.4 | 93.7 | 93.1 | 90.7 | 92.0 | 95.2 | 90.1 | 92.1 | 92.5 | 92.5 | 88.7 | 93.9 |
| SelfRegulationSCP2 | 52.8 | 52.2 | 53.3 | 54.7 | 56.3 | 60.5 | 65.6 | 50.5 | 51.1 | 55.6 | 53.3 | 56.1 | 56.0 | 56.1 | 54.4 | 60.6 |
| SpokenArabicDigits | 100.0 | 100.0 | 71.2 | 99.0 | 99.1 | 99.4 | 99.8 | 81.4 | 100.0 | 99.5 | 100.0 | 99.1 | 99.6 | 98.8 | 100 | 99.7 |
| UWaveGestureLibrary | 87.8 | 85.9 | 94.4 | 88.3 | 86.7 | 86.6 | 88.2 | 82.1 | 80.3 | 82.3 | 85.6 | 85.8 | 85.6 | 86.6 | 85.3 | 92.2 |
| **Average Accuracy** | 71.8 | 70.9 | 72.5 | 73.6 | 74.2 | 74.6 | _75.3_ | 67.5 | 70.4 | 70.9 | 72.1 | 72.5 | 73.2 | 73.0 | 70.7 | **77.6** |

# D Hyperparamter Sensitivity

In this section, we conduct a sensitivity analysis on four key hyperparameters in the SGNet model to evaluate its robustness. Specifically, we investigate the effects of (1) the number of variable groups, (2) the number of convolution kernels, (3) the selection of periodic window sizes, and (4) the embedding dimension within each group. We present a detailed evaluation of how these hyperparameters impact model performance. The corresponding experimental results are summarized below.

**Group Numbers of Varieties.** Figure 8 presents the ablation study on variable grouping. According to the number of variables in each dataset, we partition them into {1, 2, 3, 4, 5, 6} groups. Here, a group number of 1 indicates that no grouping strategy is applied and all variables are treated jointly. We observe that as the number of groups increases, performance metrics such as accuracy initially decrease and then gradually improve. Notably, the TDBRAIN dataset exhibits a significant performance gain when variable grouping is employed, highlighting the effectiveness of our method in capturing complex inter-variable relationships.

Figure 8: Analysis of hyperparameter sensitivity concerning the group numbers on main datasets.

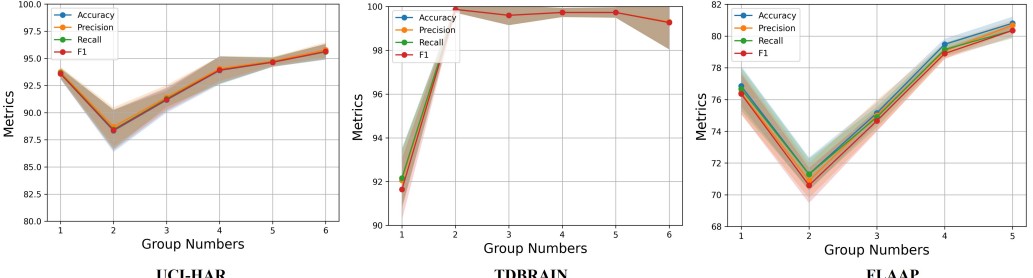

UCI-HAR          TDBRAIN          FLAAP

**Embedding Dimensions and Number of kernels.** Table 9 and Table 10 present the performance of our model under different embedding dimensions $D_m \in \{16, 32, 64, 128, 256\}$ and convolution kernel sizes $n_k \in \{3, 5, 7, 9, 11\}$, respectively. Note that the effective receptive field of each convolutional layer is $2n_k + 1$. We evaluate these configurations on the main datasets.

As shown in Table 9, the TDBRAIN dataset exhibits minimal sensitivity to changes in embedding dimension, with accuracy consistently remaining above 99%. In contrast, for the other three datasets, accuracy gradually improves as the embedding dimension increases from 16 to 64. However, further increasing the dimension beyond 64 leads to a noticeable performance drop, particularly from 128 to 256. Therefore, considering both performance and computational efficiency, embedding dimensions of 32 or 64 are identified as optimal choices. And the Table 10 shows that as the number of convolution kernels increases, leading to a larger effective receptive field, the classification accuracy improves gradually. This indicates that the model is capable of extracting temporal features effectively even with smaller kernels, demonstrating its strong robustness and reduced reliance on extensive kernel coverage.

Table 9: Analysis of hyperparameter sensitivity concerning the dataembedding on main datasets.

| Dataset | TDBrain | | PTB-XL | | FLAAP | | UCI-HAR | |
|---|---|---|---|---|---|---|---|---|
| **Metric** | Accuracy | F1 | Accuracy | F1 | Accuracy | F1 | Accuracy | F1 |
| $D_m = 16$ | 99.75 | 99.75 | 73.58 | 62.56 | 78.28 | 77.87 | 94.96 | 94.53 |
| $D_m = 32$ | 99.90 | 99.90 | 73.65 | 62.92 | 79.64 | 79.25 | 94.62 | 94.67 |
| $D_m = 64$ | 99.56 | 99.56 | 73.80 | 63.43 | 80.81 | 80.35 | 95.62 | 95.64 |
| $D_m = 128$ | 99.46 | 99.46 | 73.56 | 62.67 | 79.66 | 79.40 | 93.52 | 93.56 |
| $D_m = 256$ | 99.54 | 99.54 | 73.54 | 62.43 | 78.34 | 77.98 | 87.86 | 88.06 |

Table 10: Analysis of hyperparameter sensitivity concerning the kernel numbers on main datasets.

| Dataset | TDBrain | | PTB-XL | | FLAAP | | UCI-HAR | |
|---|---|---|---|---|---|---|---|---|
| **Metric** | Accuracy | F1 | Accuracy | F1 | Accuracy | F1 | Accuracy | F1 |
| $nk = 3$ | 99.84 | 99.84 | 73.08 | 62.44 | 80.05 | 79.45 | 94.10 | 94.08 |
| $nk = 5$ | 99.86 | 99.86 | 73.57 | 62.51 | 80.31 | 79.83 | 94.64 | 94.69 |
| $nk = 7$ | 99.90 | 99.90 | 73.80 | 63.43 | 80.81 | 80.35 | 95.62 | 95.64 |
| $nk = 9$ | 99.76 | 99.76 | 73.35 | 62.51 | 80.58 | 80.25 | 95.38 | 95.42 |
| $nk = 11$ | 99.39 | 99.39 | 73.49 | 62.73 | 80.29 | 79.86 | 94.38 | 94.45 |

**Periodic Window Sizes.** To evaluate the effectiveness of periodic window selection, we analyze the frequency-domain representations of each main dataset and select the top-5 frequencies with the highest amplitudes (excluding the full sequence length) as candidate periods. The corresponding periodic windows for each dataset are as follows: TDBRAIN (26, 128, 29, 32), FLAAP (50, 34, 25, 20), UCI-HAR (64, 32, 26, 19), and PTB-XL (84, 125, 63, 36). We then compare the model performance under these different periodic settings. As shown in Table 11, the model achieves slightly better performance under the first two period values, though the differences across configurations remain relatively small. This demonstrates the robustness of our model and validates the effectiveness of the proposed periodic mechanism for extracting multi-scale temporal information.

Table 11: Analysis of hyperparameter sensitivity concerning periodic window sizes on main datasets.

| Dataset | TDBrain | | PTB-XL | | FLAAP | | UCI-HAR | |
|---|---|---|---|---|---|---|---|---|
| **Metric** | Accuracy | F1 | Accuracy | F1 | Accuracy | F1 | Accuracy | F1 |
| $periodic\_window = No.1$ | 99.90 | 99.90 | 73.34 | 62.54 | 80.81 | 80.35 | 93.88 | 93.95 |
| $periodic\_window = No.2$ | 99.69 | 99.69 | 73.56 | 63.65 | 79.08 | 78.66 | 95.62 | 95.64 |
| $periodic\_window = No.3$ | 99.58 | 99.58 | 72.91 | 61.59 | 79.24 | 78.77 | 93.95 | 93.97 |
| $periodic\_window = No.4$ | 99.79 | 99.79 | 73.09 | 62.48 | 77.45 | 76.94 | 94.50 | 94.49 |

# E   Efficiency Analysis

We conduct a comprehensive comparison between SGN and several representative models, including Reformer [48], Crossformer [25], FEDformer [14], iTransformer [24], PatchTST [22], MICN [17], ModernTCN [19], and TimesNet [18], from the perspectives of classification performance, training speed, and memory usage. As illustrated in Figure 9, SGN achieves a well-balanced trade-off between accuracy, training time, and memory consumption on the TDBRAIN and FLAAP datasets. These results highlight the efficiency and effectiveness of our proposed method in handling multivariate time series data under resource-constrained settings.

Figure 9: Model efficiency comparison under TDBRAIN and FLAAP Datesets.

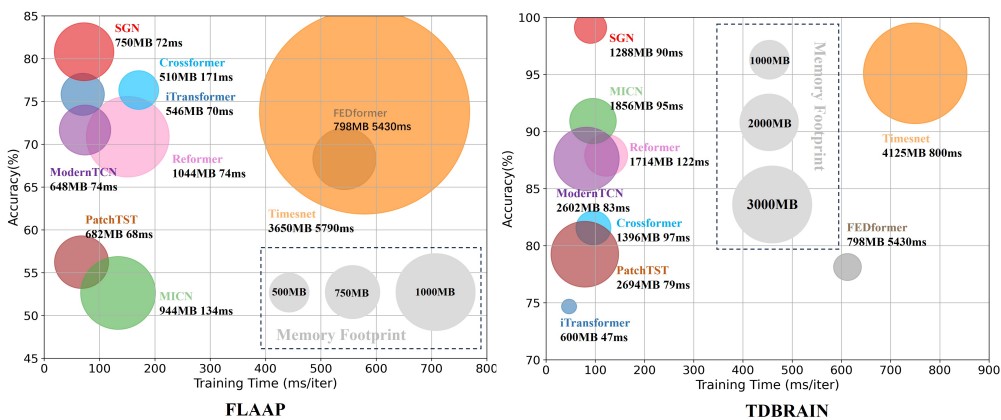

# F   Additional Visualization Results

In Figure 10, we visualize the temporal correlations across different layers of the model on various datasets. It is evident that in the lower layers, temporal dependencies are mostly confined to nearby timestamps. However, with the integration of the PWSM module, which progressively merges low-level features from short-period windows into high-level representations of long-period windows, the temporal correlations in the upper layers extend across the entire window. This demonstrates the effectiveness of our multi-level periodic fusion mechanism in capturing comprehensive global temporal patterns.

Figure 10: The temporal correlation on TDBRAIN, FLAAP and PTB-XL Datasets.

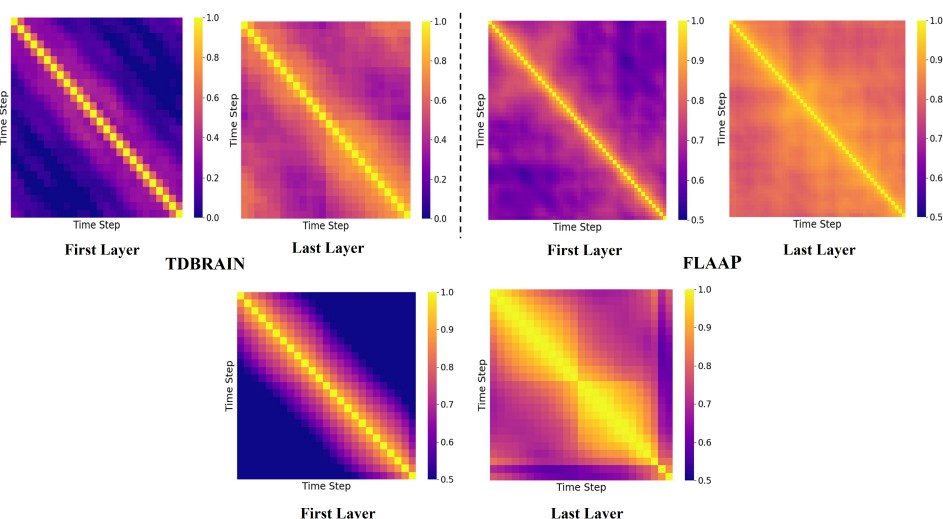

# G   Discussion

In this paper, we propose the SGN architecture, which introduces a novel perspective for modeling variable relationships by transforming them into intra-group and inter-group interactions through variable grouping. Additionally, we leverage the periodic nature of time series to perform hierarchical and multi-scale temporal feature extraction. Our method achieves state-of-the-art performance across multiple datasets from various domains, demonstrating its generalizability and effectiveness. Despite its strong empirical results and contributions to time series modeling, SGN still has certain limitations and leaves room for further exploration.

**Limitation.** While SGN demonstrates strong performance improvements and achieves outstanding results on several datasets, its applicability to other time series tasks such as forecasting and imputation remains to be further explored. Moreover, the use of intrinsic similarity to generate the assignment matrix introduces additional computational overhead. This preprocessing step, although beneficial for modeling accuracy, may hinder deployment in real-time scenarios, and thus requires further optimization and acceleration.

**Future Work.** Future research can gradually extend SGN to other domains, such as industrial monitoring and healthcare, where multivariate time series play a crucial role. In addition, exploring alternative strategies for generating the assignment matrix, potentially guided by domain knowledge, could improve the flexibility and interpretability of variable grouping. Furthermore, enhancing the design of periodic windows, including the integration of multiple window candidates or adaptive selection mechanisms, may further boost the model's effectiveness and generalization across diverse datasets.

**Broader Impacts.** The proposed SGN model holds promising potential for positive societal impact. By enhancing the accuracy and efficiency of multivariate time series classification, SGN can benefit a wide range of critical applications. For instance, in the healthcare domain, it can help detect diseases early by analyzing patient ECG signals, leading to timely interventions and improved outcomes. In meteorology, SGN can assist in making informed decisions based on complex environmental data, ultimately contributing to risk reduction and public safety. These capabilities demonstrate the model's value in supporting well-being and societal resilience across multiple sectors.

