# OpenReview forum: "SGN: Shifted Window-Based Hierarchical Variable Grouping for Multivariate Time Series Classification"
_NeurIPS.cc/2025/Conference — NeurIPS 2025 poster_

### Official Review · Reviewer_6yyx · 2025-06-18

**Clarity:** 2
**Significance:** 2
**Originality:** 2
**Rating:** 4
**Confidence:** 5

**Summary:**

This paper proposes SwinGroupNet (SGN), a novel framework for multivariate time series (MTS) classification that addresses limitations of existing methods by integrating structured variable grouping and hierarchical temporal feature extraction. The model comprises three core modules: 1) Variable Group Embedding (VGE), which partitions variables into groups based on similarity to capture intra-group and inter-group dependencies; 2) Multi-Scale Group Window Mixing (MGWM), which extracts multi-scale temporal features via periodic window partitioning and convolutional operations; 3) Periodic Window Shifting and Merging (PWSM), which leverages sequence periodic patterns to enable hierarchical temporal interaction. Extensive experiments on diverse benchmarks show SGN achieves state-of-the-art performance, with an average 4.2% accuracy improvement over existing methods.

**Questions:**

--- In the Variable Group Embedding module, BDC (Brownian Distance Covariance) is used to model dependencies between variables. However, referring to Equation (11), its essence still relies on Euclidean distance. Does this approach genuinely capture better non-linear dependencies? Is there any performance improvement compared to other methods? Additionally, when calculating dependencies between variables, do you use the entire time series of that variable?
--- "Considering the extensibility of periodic patterns, we further incorporate multiple scales by merging additional period windows corresponding to the remaining Top-K dominant frequencies." (page=5, lines 155-157) This statement mentions merging period windows corresponding to the remaining frequencies in Top-K. Could you elaborate in detail on how this merging process is implemented?
--- While moving and merging periodic windows enhances the capture of cross-window dependencies, what specific offset value and window size after merging yield optimal results?
--- In Section 4.2, the authors selected 10 multivariate time series datasets from the UEA. Many research reports have shown the performance of all datasets. How does your work perform on all datasets? Additionally, the hyperparameter sensitivity analysis lacks an evaluation of the regularization parameter $\beta$.

**Ethical Concerns:**

["NO or VERY MINOR ethics concerns only"]

**Final Justification:**

I have increased my rating considering the work from authors to clarify and improve the paper.

**Limitations:**

The applicability of SGN in other time series tasks such as forecasting and imputation remains to be further explored. Moreover, the use of intrinsic similarity to generate the assignment matrix introduces additional computational overhead.

**Paper Formatting Concerns:**

None.

**Quality:**

2

**Strengths And Weaknesses:**

Strengths:
1. This paper introduces a novel perspective transforming multivariate interactions into structured intra/inter-group relationships, effectively balancing fine-grained dependency capture and over-smoothing avoidance.
2. Experimental validation covers domains including healthcare and human activity recognition, demonstrating consistent superiority over baselines.

Weaknesses:
1. In the Variable Group Embedding module, BDC (Brownian Distance Covariance) is used to model dependencies between variables. However, referring to Equation (11), its essence still relies on Euclidean distance. Does this approach genuinely capture better non-linear dependencies? Is there any performance improvement compared to other methods? Additionally, when calculating dependencies between variables, do you use the entire time series of that variable?
2. "Considering the extensibility of periodic patterns, we further incorporate multiple scales by merging additional period windows corresponding to the remaining Top-K dominant frequencies." (page=5, lines 155-157) This statement mentions merging period windows corresponding to the remaining frequencies in Top-K. Could you elaborate in detail on how this merging process is implemented?
3. In the "Multi-Scale Group Window Extracting" section (Lines 159–172), operations of intra-group and inter-group pointwise convolutions are not explained. Mathematical formulations are needed to clarify channel dimension transformations.
4. While moving and merging periodic windows enhances the capture of cross-window dependencies, what specific offset value and window size after merging yield optimal results? Please provide a quantitative analysis of this.
5. In Section 4.2, the authors selected 10 multivariate time series datasets from the UEA. Many research reports have shown the performance of all datasets. How does your work perform on all datasets? Additionally, the hyperparameter sensitivity analysis lacks an evaluation of the regularization parameter $\beta$.
6. Notation inconsistency: $k_i$ in Equation (6) conflicts with $n_k$ in Line 699 of Appendix D, causing confusion. Notation should be unified.

---

> ### Author Rebuttal · Authors · 2025-07-31
>
> We sincerely appreciate your recognition of our novel perspective on modeling multivariate interactions through structured intra/inter-group relationships, as well as your acknowledgment of the strong empirical validation across diverse domains. We would like to provide some clarifications in the hope of addressing your concerns.
>
> ---
>
>
> **W1. While the VGE module employs BDC to model inter-variable dependencies, Equation (11) reveals its reliance on Euclidean distance, raising questions about its ability to truly capture non-linear relationships. Clarification is needed on whether this offers performance gains over other dependency measures, and whether the full time series of each variable is used in the computation.**
>
>
> **A1**. We thank the reviewer for the insightful question regarding our similarity metric.
> In our method, we adopt BDC to measure similarity, where a positive value of **dCov(X, Y) > 0** indicates the existence of statistical dependence—including **non-linear** relationships.
> To validate its effectiveness, we compare **BDC** with **Pearson correlation** on both the FLAAP and UCI-HAR datasets. As shown in the table below, BDC demonstrates a **slight improvement in performance**.
> Notably, BDC computes similarity over the **entire time series**, allowing it to better preserve the temporal structure during the similarity estimation process.
>
> | Dataset |  FLAAP   |       | UCI-HAR  |       |
> | ------- | :------: | :---: | :------: | :---: |
> | Metric  | Accuracy |  F1   | Accuracy |  F1   |
> | Pearson |  79.64   | 78.94 |  95.52   | 95.55 |
> | BDC     |  80.81   | 80.35 |  95.62   | 95.64 |
>
> ---
>
> **W2. The paper mentions merging period windows from the remaining Top-K frequencies, but it lacks details—could the authors clarify how this merging is implemented in practice?**
>
> **A2.**  In our experiments, we observed that the **top-k dominant periods** in time series often exhibit a **multiple relationship**. Based on this insight, we design our model to start with a **small base period window**, and then progressively **fuse multiple windows** to reach larger period lengths.
> This periodic window fusion strategy allows SGN to capture diverse periodic patterns effectively across various temporal resolutions.
> ****
>
> **W3. In the "Multi-Scale Group Window Extracting" section (Lines 159–172), operations of intra-group and inter-group pointwise convolutions are not explained. Mathematical formulations are needed to clarify channel dimension transformations.**
>
> **A3.** Given the input data tensor **$X \in \mathbb{R}^{num \times (d_{\text{model}} \times v_{\text{group}}) \times T}$**, where `num` is the number of temporal windows, `T` is the periodic window length, $v_{\text{group}}$​​ is the number of variable groups, and $d_{\text{model}}$​​ is the feature dimension per group:
>
> - We first apply **group-wise pointwise convolutions** by setting the number of convolution groups to $v_{\text{group}}$​​, enabling **intra-group interaction**.
> - Next, we **transpose the $v_{\text{group}}$ and $d_{\text{model}}$​​ dimensions**, resulting in a shape of **num × ($v_{\text{group}}$​​ x $d_{\text{model}}$​​) × T**.
> - Then, we apply another pointwise convolution with **$d_{\text{model}}$​​** groups, allowing **inter-group feature interactions** across different variable embeddings.
>
> This two-stage group-wise and inter-group convolution design enhances both **local specialization** and **global coordination** across variable groups.
>
> ---
>
> **Q4. What are the specific offset and merged window size values that optimize cross-window dependency capture, and could you provide quantitative analysis supporting these choices?**
>
> **A4.** We provide detailed ablation results on the effect of **shift offsets** according to the window length T in the table below. As shown, different offset values yield similar performance, indicating that the essential contribution lies in enabling **information exchange across shifted temporal windows**, rather than relying on a specific offset.
>
> | Shift_offset  |   0   |  T/4  |  T/2  | 3/4T  |
> | ------- | :---: | :---: | :---: | :---: |
> | FLAAP   | 78.05 | 80.07 | 80.81 | 79.48 |
> | UCI-HAR | 87.05 | 94.74 | 95.62 | 94.62 |
> | TDBRAIN | 99.79 | 99.91 | 99.9  | 99.75 |
> | PTB-XL  | 73.32 | 73.75 | 73.8  | 74.19 |
>
> For the **window merging** strategy, we adopt a doubling-based merging approach, where the merged window size is a multiple of the original. The corresponding results and in-depth analysis are presented in **Table 10 of Appendix E**, demonstrating the effectiveness of our period merging design.
>
>
> ---
> **Q5. The evaluation on only 10 UEA datasets is limited; how does the method perform on the full set? Also, the hyperparameter sensitivity analysis omits the regularization parameter.**
>
> **A5.** We have supplemented the results on the full UEA multivariate time series classification benchmark. Under the **same experimental settings**, we compare SGN with a broader range of recent state-of-the-art methods (see the Table). **Wins/Draws/Losses** indicate the number of datasets (out of 30) on which the SGN method achieves higher, equal, or lower accuracy compared to the corresponding baseline methods. As shown, SGN consistently achieves **a greater number of first-place results and better average ranking** compared to state-of-the-art methods, demonstrating its robustness and generalization across diverse datasets.
>
> |Data/Model|W.MUSE|M.FCN|TapNet|ShapeNet|TodyNet|SVPT|Shapeformer|MPTSNet|SGN(Ours)|
> |---|:--:|:--:|:--:|:--:|:--:|:--:|:--:|:--:|:--:|
> |ArticularyWordRecognition|99|97.3|98.7|98.7|98.7|**99.3**|99|97.7|99|
> |AtrialFibrillation|33.3|26.7|33.3|40|46.7|40|53.3|53.3|**66.7**|
> |BasicMotions|**100**|95|**100**|**100**|**100**|**100**|**100**|**100**|**100**|
> |CharacterTrajectories|99|98.5|**99.7**|98|N/A|99|99.2|N/A|99.6|
> |Cricket|**100**|91.7|95.8|98.6|**100**|**100**|94.4|94.4|**100**|
> |DuckDuckGeese|57.5|67.5|57.5|**72.5**|58|70|64|68|64|
> |EigenWorms|89|50.4|48.9|87.8|84|**92.5**|N/A|N/A|85.5|
> |Epilepsy|**100**|76.1|97.1|98.7|97.1|98.6|98.6|97.1|97.8|
> |EthanolConcentration|13.3|37.3|32.3|31.2|35|33.1|41.1|43.3|**44.5**|
> |ERing|43|13.3|13.3|13.3|91.5|93.7|87.4|94.4|**95.9**|
> |FaceDetection|54.5|54.5|55.6|60.2|62.7|51.2|65.8|69.8|**70.3**|
> |FingerMovements|49|58|53|58.9|**67.6**|60|55|64|64|
> |HandMovementDirection|36.5|36.5|37.8|33.8|64.9|39.2|41.9|63.5|**75.7**|
> |Handwriting|**60.5**|28.6|35.7|45.1|43.6|43.3|30.2|34.4|50.4|
> |Heartbeat|72.7|66.3|75.1|75.6|75.6|79|**81.5**|75.6|77.1|
> |InsectWingbeat|N/A|16.7|20.8|25|N/A|18.4|31.4|N/A|**68**|
> |JapaneseVowels|97.3|97.6|96.5|98.4|N/A|97.8|**99.2**|98.6|98.9|
> |Libras|87.8|85.6|85|85.6|85|88.3|**95.5**|87.2|83.9|
> |LSST|59|37.3|56.8|59|61.5|**66.6**|63.8|60.4|63.7|
> |MotorImagery|50|51|59|61|64|**65**|N/A|**65**|**65**|
> |NATOPS|87|88.9|93.9|88.3|97.2|90.6|96.1|94.4|**98.3**|
> |PenDigits|94.8|97.8|98|97.7|98.7|98.3|**99.1**|98.9|**99.1**|
> |PEMS-SF|N/A|69.9|75.1|75.1|78|86.7|N/A|**94.2**|88.4|
> |PhonemeSpectra|19|11|17.5|29.8|**30.9**|17.6|29.3|14.4|23.1|
> |RacketSports|**93.4**|80.3|86.8|88.2|80.3|84.2|88.8|87.5|**93.4**|
> |SelfRegulationSCP1|71|87.4|65.2|78.2|89.8|88.4|91.8|92.8|**93.9**|
> |SelfRegulationSCP2|46|47.2|55|57.8|55|60|56.1|57.2|**60.6**|
> |SpokenArabicDigits|98.2|99|98.3|97.5|N/A|98.6|**99.7**|99.5|**99.7**|
> |StandWalkJump|33.3|6.7|40|53.3|46.7|46.7|**66.7**|53.3|53.3|
> |UWaveGestureLibrary|91.6|89.1|89.4|90.6|85|**94.1**|90|88.1|92.2|
> |Average rank|5.57|7.17|6.13|4.80|4.27|3.70|3.19|3.74|**2.13**|
> |Number of top-1|5|0|2|2|4|7|7|3|**15**|
> |Wins|20|28|27|23|21|19|15|20|-|
> |Draws|4|0|1|2|2|3|5|4|-|
> |Loses|4|2|2|5|3|8|7|3|-|
>
> In addition, we conduct an ablation study on the **β parameter** (see Table Y). The results reveal a **non-monotonic trend**: as β increases, the performance initially improves and then gradually declines, indicating that an appropriate choice of β is crucial for balancing the similarity regularization term.
>
> | β       |   0   |  0.1  |  0.2  |  0.3  |
> | ------- | :---: | :---: | :---: | :---: |
> | FLAAP   | 80.24 | 80.81 | 79.4  | 80.09 |
> | UCI-HAR | 94.86 | 95.62 | 94.05 | 94.42 |
> | TDBRAIN | 99.88 | 99.9  | 99.75 | 99.88 |
> | PTB-XL  | 70.84 | 73.8  | 73.84 | 73.8  |
>
> ---
>
> **W6. Notation inconsistency:  in Equation (6) conflicts with  in Line 699 of Appendix D, causing confusion. Notation should be unified**
>
> **A6.** We denote $k_{\text{i}}$​​ as the kernel size and $n_{\text{k}}$​​​ as the number of convolution kernels. For example, when $n_{\text{k}}$​​=6, we use multiple convolution kernels with sizes $k_{\text{i}}$​​∈{1,3,5,7,9,11}. This design enables the model to capture temporal patterns at multiple scales effectively.
>
>
> ---
>
> **Q. The problem is the same as the weakness.**
>
> **A7.** We have provided the corresponding responses in **Answers A1–A6** above.

---

> ### Author Response · Authors · 2025-08-06
>
> Dear Reviewer,
>
> We would like to kindly follow up regarding our rebuttal. We value your feedback greatly and would appreciate it if you could share any additional comments or questions when convenient. We are happy to provide further clarifications, experiments, or supporting materials at any time to facilitate the discussion.
>
> Thank you very much for your time and consideration.

---

### Official Review · Reviewer_eNCr · 2025-06-29

**Clarity:** 3
**Significance:** 2
**Originality:** 2
**Rating:** 4
**Confidence:** 5

**Summary:**

This paper proposes SwinGroupNet (SGN), a novel framework for multivariate time series classification, which integrates three key components: Variable Group Embedding (VGE), Multi-scale Group Window Mixing (MGWM), and Periodic Window Shifting and Merging (PWSM). By combining these modules, SGN effectively captures both intra-group and inter-group variable dependencies across multiple temporal scales from raw multivariate time series data. Extensive experiments on four medical multivariate time series datasets and ten benchmark datasets from the UEA archive demonstrate that SGN achieves state-of-the-art classification performance.

**Questions:**

1. The UEA archive does not provide predefined validation sets for its sub-datasets. However, in the provided source code (data_loader.py, lines 723–725), it appears that a validation set is being generated. Could the authors clarify the strategy used to create the validation sets for the UEA datasets? Additionally, the released code does not include experimental scripts for running on the UEA datasets—would the authors consider making them publicly available to ensure reproducibility?

2. Regarding the results shown in Figure 4, were all baseline results (e.g., ROCKET, TimesNet) re-implemented and re-evaluated by the authors under the same experimental setup? If so, could the authors provide details on how the reproduction was performed, including any modifications or hyperparameter settings used?

3. Given the experimental settings adopted in [R8, R9] on the full UEA 30 time series dataset, could the authors report the classification performance of SGN under the same setting, using the published results from [R8, R9] as baselines (without the need to re-run those methods if time and resources are limited)? Such a comparison would provide a clearer understanding of SGN’s effectiveness relative to recent state-of-the-art approaches.

4. The weaknesses outlined above—including concerns about incremental novelty, unclear interactions among proposed modules, limited UEA dataset coverage, and insufficient comparison with recent related methods—require further clarification and justification.


[R9] MPTSNet: Integrating multiscale periodic local patterns and global dependencies for multivariate time series classification. AAAI, 2025.

**Ethical Concerns:**

["NO or VERY MINOR ethics concerns only"]

**Final Justification:**

Based on the author's clarification,  I have decided to raise my score to Borderline Accept.

**Limitations:**

Yes.

**Paper Formatting Concerns:**

None.

**Quality:**

3

**Strengths And Weaknesses:**

Strengths：

1. The proposed VGE, MGWM, and PWSM modules are thoughtfully designed to address key aspects of multivariate time series analysis, including variable-wise dependency modeling, multi-scale feature extraction, and frequency-domain periodicity—factors that are central to advancing current time series research.

2. The figures and tables are clearly presented and enhance the interpretability of the proposed method.

3. The related work section is well-organized and comprehensive, providing a clear overview of the field’s recent advancements.

Weaknesses：

1. The overall novelty of the proposed SGN appears incremental, as it primarily integrates ideas from existing works. For instance, the core concept of the VGE module has been previously summarized and discussed in [R1], and shares similarities with contributions from [R2, R3]. Likewise, the multi-scale mixing strategy in MGWM resembles the approach adopted in TimeMixer [R4].

2. Although VGE, MGWM, and PWSM are individually designed to capture distinct characteristics of multivariate time series, the conceptual and functional relationships among them are not clearly articulated. In particular, MGWM already captures multi-scale temporal patterns, which often include periodic structures. It remains unclear how the additional periodic modeling via PWSM complements or interacts with MGWM, and what underlying mechanism ensures their synergy.

3. The authors claim in the abstract that existing methods either ignore inter-variable dependencies or model all variables jointly. However, a number of prior works have explicitly addressed inter-variable dependency modeling in multivariate time series classification [R5, R6], which are not discussed or empirically compared in the paper.

4. For multivariate time series classification, the UEA 30 dataset is a widely accepted benchmark [R7, R8]. The authors evaluate their method on only 10 selected datasets, raising concerns about potential cherry-picking and limited generalizability of the results.


[R1] A Comprehensive Survey of Deep Learning for Multivariate Time Series Forecasting: A Channel Strategy Perspective. arXiv, 2025.

[R2] Crossformer: Transformer utilizing cross-dimension dependency for multivariate time series forecasting. ICLR, 2023.

[R3] From similarity to superiority: Channel clustering for time series forecasting. NeurIPS, 2024.

[R4] Timemixer: Decomposable multiscale mixing for time series forecasting. ICLR, 2024.

[R5] SVP-T: A shape-level variable-position transformer for multivariate time series classification. AAAI, 2023.

[R6] Fully-Connected Spatial-Temporal Graph for Multivariate Time Series Data. AAAI, 2024.

[R7] The great multivariate time series classification bake off: a review and experimental evaluation of recent algorithmic advances. DMKD, 2021.

[R8] Shapeformer: Shapelet transformer for multivariate time series classification. KDD, 2024.

---

> ### Author Rebuttal · Authors · 2025-07-31
>
> Many thanks for your thoughtful and encouraging comments. We sincerely appreciate your recognition of the design and motivation behind our proposed VGE, MGWM, and PWSM modules. We would like to offer further clarifications to enhance the understanding of our contributions.
>
> ---
> **W1. The novelty of SGN seems incremental, as its core components (VGE, multi-scale mixing) closely resemble ideas from prior works such as [R1–R3].**
>
> **A1**. We would like to clarify the design and motivation behind our modeling of variable dependencies. We have observed in our experiments that performing variable aggregation based solely on intrinsic similarity at the **initial stage** fails to capture the **complex dependencies** among variables (see the ablation results in Table 3). To address this, we propose a **two-stage variable dependency modeling strategy** in SGN.
>
> In the **first stage**, **VGE** module leverages pairwise similarity to group variables and perform group embedding. However, our goal here is to obtain a **coarse segmentation of variables**, analogous to patching along the temporal axis, allowing for more efficient downstream processing. In the **second stage**, we explicitly model **major variable dependencies** within and across variable groups. This allows for deeper and more expressive modeling of intra- and inter-group variable relationships.
>
> Compared with existing methods:
>
> - [R1] treats variable dependency modeling as a **plugin aggregation module** and does not further explore dependencies after grouping. In contrast, SGN performs **continued hierarchical interactions** after the initial grouping.
>
> - [R2] applies operations to all variables **individually**, without leveraging variable grouping as an inductive bias. This may lead to **over-smoothing** and less structured dependency modeling.
>
> Regarding **multi-scale strategies**, [R3] applies linear projections to generate multiple temporal views. In contrast, **SGN leverages the strong local feature extraction capability of convolutions** and employs multiple 1D convolutional kernels with different receptive fields to extract **multi-scale patterns** within periodic windows.
>
> [R1] From similarity to superiority: Channel clustering for time series forecasting. NeurIPS, 2024.
>
> [R2] Crossformer: Transformer utilizing cross-dimension dependency for multivariate time series forecasting. ICLR, 2023.
>
> [R3] Timemixer: Decomposable multiscale mixing for time series forecasting. ICLR, 2024.
>
> ---
> **W2. The conceptual distinctions and interactions among VGE, MGWM, and PWSM are unclear, particularly regarding how PWSM complements MGWM’s existing multi-scale temporal modeling.**
>
> **A2**. We appreciate the opportunity to further clarify the **interactions among SGN’s core modules**.
>
> - The **VGE module** serves as the foundation by constructing both the **periodic windows** and **variable groups**, enabling structured and localized modeling.
>
> - Built upon the VGE outputs, the **MGWM module** operates **within each periodic window**, performing **feature extraction along both the temporal and variable dimensions**.
>
> - While MGWM focuses on **within-window modeling**, PWSM is designed to **capture dependencies across windows**, thereby enabling global information flow and alignment across different temporal windows.
>
> These three components are **mutually supportive and closely coupled**.
>
> ---
> **W3. The paper overlooks prior works that explicitly model inter-variable dependencies, despite claiming a gap in existing methods without proper discussion or comparison.**
>
> **A3**. We divide variable interactions into three main categories: **variable-independent modeling**, **variable aggregation**, and **variable mixing**. These three strategies have been explicitly discussed in the _Related Work_ section to highlight their conceptual differences and practical trade-offs. We appreciate the reviewer’s attention to this aspect and **will incorporate additional relevant literature** in the final version to provide a more comprehensive overview of related approaches.
>
> ---
> **W4. Evaluating on only 10 selected datasets from the UEA archive, rather than the full 30, raises concerns about cherry-picking and limits the generalizability of the results.**
>
> **A4**. We evaluate the effectiveness of SGN on four major datasets. To further assess the robustness and generalizability of SGN under diverse datasets—such as long sequences and weak periodicity—we additionally conduct experiments on ten UEA multivariate time series datasets. These datasets are commonly used in prior studies such as [R4-R6]. We have added results on all 30 UEA datasets and compared SGN with more baseline methods, as summarized in the table below.
>
> [R4] TimesNet: Temporal 2D-Variation Modeling for General Time Series Analysis. ICLR, 2023.
>
> [R5] ModernTCN: A Modern Pure Convolution Structure for General Time Series Analysis. ICLR, 2024.
>
> [R6] TimeMixer++: A General Time Series Pattern Machine for Universal Predictive Analysis. ICLR, 2025.
>
> ---
> **Q1. The paper lacks clarity on the validation set construction for UEA datasets, as the code suggests a custom split despite no predefined sets. Moreover, the absence of experimental scripts for UEA datasets hinders reproducibility.**
>
> **A5.** We adopt a fixed random seed and utilize the **StratifiedShuffleSplit** method to ensure that the class distribution in the split subsets remains consistent with the original dataset. Specifically, we reserve **20% of the training set as the validation set**, and use the remaining **80% for model training**.
> Due to NIPS policy, we are unable to upload our UEA implementation script to the Anonymity link. However, we will make the code publicly available after the review process to ensure full reproducibility.
>
> ---
> **Q2. Figure 4 lacks clarity on whether baselines were re-implemented under a consistent setup; details on reproduction, modifications, and hyperparameter settings are needed.**
>
> **A6.** We directly follow the data used in their paper. However, unlike their setting where the **test set is directly used as a validation set**, we adopt a more rigorous strategy by **reserving 20% of the training set as a validation set**, which helps prevent information leakage and better simulates real-world scenarios.
>
> ---
> **Q3. To better assess SGN’s effectiveness, the authors are encouraged to report its performance under the same experimental settings used in [R7, R8] on the full UEA 30 dataset, using published results as baselines if re-running those methods is impractical.**
>
> **A7.** To further validate the effectiveness of the SGN method, we have supplemented additional comparison results under the same experimental settings with methods such as **Shapeformer** and **MPTSNet** on the **full UEA dataset**.
> Due to time constraints, we were only able to reproduce a subset of the existing methods. The detailed results are provided in the following table. As shown, SGN consistently achieves **a greater number of first-place results and better average ranking** compared to state-of-the-art methods, demonstrating its robustness and generalization across diverse datasets.
> **Notably**, the **SGN#** column reports results using 20% of the training set as a validation set, **rather than directly using the test set for training as in other settings**, and still demonstrates strong performance.
>
> |Data/Model|W.MUSE|M.FCN|TapNet|ShapeNet|TodyNet|SVPT|Shapeformer|MPTSNet|SGN|SGN#|
> |--|:-:|:-:|:-:|:-:|:-:|:-:|:-:|:-:|:-:|:-:|
> |AW|99|97.3|98.7|98.7|98.7|**99.3**|99|97.7|99|98|
> |AF|33.3|26.7|33.3|40|46.7|40|53.3|53.3|**66.7**|53.3|
> |BasicMotions|**100**|95|**100**|**100**|**100**|**100**|**100**|**100**|**100**|**100**|
> |Character|99|98.5|**99.7**|98|N/A|99|99.2|N/A|99.6|97.8|
> |Cricket|**100**|91.7|95.8|98.6|**100**|**100**|94.4|94.4|**100**|98.6|
> |DDG|57.5|67.5|57.5|**72.5**|58|70|64|68|64|62|
> |EWorms|89|50.4|48.9|87.8|84|**92.5**|N/A|N/A|85.5|80.2|
> |Epilepsy|**100**|76.1|97.1|98.7|97.1|98.6|98.6|97.1|97.8|97.1|
> |EC|13.3|37.3|32.3|31.2|35|33.1|41.1|43.3|**44.5**|**44.5**|
> |ERing|43|13.3|13.3|13.3|91.5|93.7|87.4|94.4|**95.9**|91.9|
> |FaceDetection|54.5|54.5|55.6|60.2|62.7|51.2|65.8|69.8|**70.3**|**70.3**|
> |FingerMove|49|58|53|58.9|**67.6**|60|55|64|64|57|
> |HandMove|36.5|36.5|37.8|33.8|64.9|39.2|41.9|63.5|**75.7**|60.8|
> |Handwriting|**60.5**|28.6|35.7|45.1|43.6|43.3|30.2|34.4|50.4|43.2|
> |Heartbeat|72.7|66.3|75.1|75.6|75.6|79|**81.5**|75.6|77.1|77.1|
> |Insect|N/A|16.7|20.8|25|N/A|18.4|31.4|N/A|**68**|66.2|
> |JV|97.3|97.6|96.5|98.4|N/A|97.8|**99.2**|98.6|98.9|98.9|
> |Libras|87.8|85.6|85|85.6|85|88.3|**95.5**|87.2|83.9|83.9|
> |LSST|59|37.3|56.8|59|61.5|**66.6**|63.8|60.4|63.7|60.8|
> |MI|50|51|59|61|64|**65**|N/A|**65**|**65**|57|
> |NATOPS|87|88.9|93.9|88.3|97.2|90.6|96.1|94.4|**98.3**|95.6|
> |PenDigits|94.8|97.8|98|97.7|98.7|98.3|**99.1**|98.9|**99.1**|98.7|
> |PEMS-SF|N/A|69.9|75.1|75.1|78|86.7|N/A|**94.2**|88.4|88.4|
> |Phoneme|19|11|17.5|29.8|**30.9**|17.6|29.3|14.4|23.1|20.4|
> |RS|**93.4**|80.3|86.8|88.2|80.3|84.2|88.8|87.5|**93.4**|91.4|
> |SCP1|71|87.4|65.2|78.2|89.8|88.4|91.8|92.8|**93.9**|91.1|
> |SCP2|46|47.2|55|57.8|55|60|56.1|57.2|**60.6**|57.8|
> |SA|98.2|99|98.3|97.5|N/A|98.6|**99.7**|99.5|**99.7**|**99.7**|
> |StandWalkJump|33.3|6.7|40|53.3|46.7|46.7|**66.7**|53.3|53.3|40|
> |UWave|91.6|89.1|89.4|90.6|85|**94.1**|90|88.1|92.2|90.3|
> |Average rank|6.21|8.03|6.13|5.30|4.69|4.13|3.56|4.30|**2.13**|4.40|
> |Number of top-1|5|0|2|2|4|7|7|3|**15**|4|
> |Wins|20|28|27|23|21|19|15|20|-||
> |Draws|4|0|1|2|2|3|5|4|-||
> |Loses|4|2|2|5|3|8|7|3|-||
>
> [R7] MPTSNet: Integrating multiscale periodic local patterns and global dependencies for multivariate time series classification. AAAI, 2025.
>
> [R8] Shapeformer: Shapelet transformer for multivariate time series classification. KDD, 2024.
>
> ---
>
> **Q4. The weaknesses outlined above require further clarification and justification.**
>
> **A8.** We have provided the corresponding responses in **Answers A1–A4** above.

---

> > ### Comment · Reviewer_Tx33 · 2025-08-04
> >
> > The authors addressed my concerns.

---

> ### Comment · Reviewer_eNCr · 2025-08-04
>
> **Regarding W1 and W2**: The authors' responses have addressed most of my concerns. However, I still believe the methodological novelty is incremental.
>
> **Regarding W3**: My main concern is that the paper lacks in-depth analysis of works on multivariate time series *classification*. The discussion on variable-independent modeling, variable aggregation, and variable mixing in the Related Work section primarily focuses on *forecasting* tasks. However, the title and experiments in this paper are classification-oriented. Furthermore, the authors do not sufficiently review related work specifically on multivariate time series classification.
>
> **Regarding W4**: The authors state that the selected datasets are commonly used in prior works such as \[R4–R6]. However, \[1] points out (page 8) that many recent SOTA baselines (e.g., those shown in Figure 4 of the main text) suffer from leakage—using the test set as the validation set. If all deep learning models follow this setup, comparisons may be fair among them. However, Rocket is a non-deep learning method and does not rely on such validation practices, making the comparison unfair.
>
> **Regarding Q1**: The authors clarify that they “reserve 20% of the training set as the validation set, and use the remaining 80% for model training.” However, their results in Figure 4 outperform the best baseline by 1.2%. According to \[1], if other baselines use the test set for validation, how can SGN—with stricter validation from training data only—achieve significantly better performance?
>
> **Regarding Q2**: While I appreciate the authors' clarification, I still question the rationale for including comparison results in Figure 4 that are based on validation leakage. I suggest referencing \[1] and clearly distinguishing experimental setups.
>
> **Regarding Q3**: MPTSNet follows the settings of \[R4–R6], achieving an average accuracy of 75.4% on 10 UEA datasets. The authors report an average accuracy of 76.1% in Figure 4. Meanwhile, MPTSNet, using strict training-validation splits (i.e., not using the test set, if it is true), reports 74.0% average accuracy over 25 UEA datasets.
>
> From the SGN results provided, the paper reports an average accuracy of 76.81% on 25 UEA datasets. SGN# (using 20% of training set as validation) achieves 73.17%, which is similar to Shapeformer (73.45%). This raises the question: How does SGN achieve 76.81% under the UEA-25 setting when MPTSNet, under the same setting, only reaches 74.0%?
>
> Overall, the discrepancy between experimental setups—especially regarding validation leakage—needs to be thoroughly clarified. This is particularly important for the results shown in Figure 4. If the authors can clearly address these issues in the final version, I will consider raising my score.
>
> **Reference**:
> \[1] *TOTEM: TOkenized Time Series Embeddings for General Time Series Analysis*, TMLR, 2024.

---

> > ### Author Response · Authors · 2025-08-04
> >
> > We sincerely appreciate your insightful suggestions. We will incorporate additional clarifications and explanations in the revised version of the paper.
> >
> > ---
> > **Response to W3:**
> >
> > Our current paper primarily focuses on methods that approach the problem from the **variable-wise perspective**. In future versions, we will incorporate a broader set of related works on MTSC to provide a more comprehensive context.
> >
> > ---
> >
> > **Response to W4/Q1/Q2:**
> >
> > We sincerely thank the reviewer for raising this important concern regarding validation leakage in prior works.
> >
> > In our experiments reported in **Figure 4**, we adopt a stricter protocol by reserving **20% of the training set as the validation set**, while using the remaining 80% for model training. Under this setup, SGN achieves a top-1 accuracy of **76.1%**, already outperforming other baselines by **1.2%**, despite the stricter validation policy.
> >
> > To ensure **fair and comprehensive comparisons**, we will additionally report SGN’s performance using the same protocol adopted by other baselines (i.e., using the test set as the validation set), which yields a slightly higher accuracy of **77.6%**. This updated result will be included in the **final version** of the paper along with clear annotations and descriptions distinguishing the experimental setups.
> >
> > We emphasize that SGN’s strong performance stems not from validation policies but from its core architectural design: the **VGE module** constructs variable groups and periodic windows to expose inherent structure; the **MGWM module** jointly models temporal and inter-variable interactions within each periodic window using multi-scale convolutions; and the **PWSM module** captures rich dependencies **between windows**. These three modules work synergistically to enable robust and generalizable modeling of multivariate time series, which is consistently reflected in our empirical results across diverse benchmarks.
> >
> > We appreciate the reviewer’s suggestion to reference [R1], and we will cite it appropriately and further clarify all experimental protocols in the final version.
> >
> > ---
> > **Response to Q3:**
> >
> > We would like to further clarify the experimental protocol. Under the same evaluation setting as adopted by prior works [R2–R4]—where the **test set is used for validation**—SGN achieves an average accuracy of **77.6%** on the 10 UEA datasets, compared to **75.4%** for MPTSNet. In contrast, the result reported in the main paper (76.1%) was obtained using **20% of the training set as the validation set**, which is a stricter and more realistic setup.
> >
> > Furthermore, for the **25 UEA datasets** we report in the Rebuttal, SGN achieves **76.81%** average accuracy under the same experimental setting (i.e., using the test set for validation), while MPTSNet achieves **74.0%**. This setup is consistent with the implementation in MPTSNet’s public codebase on github(see `data_provide.py`, line 69 and `train.py`, line 152, where `test_loader` is used for validation).
> > In contrast, **SGN#** refers to our result using **20% of the training set for validation**, maintaining a stricter evaluation protocol. We also include results under both validation setups in the updated tables for completeness.
> >
> > | Datasets | Model/setting | 20% train dataset for valid | test dataset for valid |
> > | -------- | ------------- | --------------------------- | ---------------------- |
> > | UEA-10   | SGN           | 76.1                        | 77.6                   |
> > |          | MPTSNet       | /                           | 75.4                   |
> > | UEA-25   | SGN           | 73.17                       | 76.81                  |
> > |          | MPTSNet       | /                           | 74.00                  |
> >
> > The superior performance of SGN can be attributed to its effective modeling of both **variable-wise** and **temporal-wise** dependencies. SGN explicitly captures intra- and inter-group variable interactions and performs multi-scale temporal extraction. In contrast, MPTS primarily focuses on temporal patterns and lacks dedicated modeling on the variable dimension, which may limit its performance on complex multivariate data.
> >
> > To ensure **reproducibility**, we will release our full UEA experiment scripts, and provide clear documentation on the experimental settings and configurations.
> >
> > [R1] TOTEM: TOkenized Time Series Embeddings for General Time Series Analysis_, TMLR, 2024.
> >
> > [R2] TimesNet: Temporal 2D-Variation Modeling for General Time Series Analysis. ICLR, 2023.
> >
> > [R3] ModernTCN: A Modern Pure Convolution Structure for General Time Series Analysis. ICLR, 2024.
> >
> > [R4] TimeMixer++: A General Time Series Pattern Machine for Universal Predictive Analysis. ICLR, 2025.

---

> > > ### Comment · Reviewer_eNCr · 2025-08-04
> > >
> > > Thank you for your response. Based on your clarification, I now understand that MPTSNet reports classification results on the UEA 25 datasets using the test set as the validation set. This experimental setup is unfair to deep learning methods like Shapeformer and non-deep learning baselines such as Rocket and DTW, which do not use test samples during training.
> > >
> > > According to your reply, under a fair setting where 20% of the training set is used as the validation set (e.g., SAN# with an average accuracy of 73.17%), your proposed method performs on par with existing SOTA methods such as Shapeformer (average accuracy of 73.45%) on the UEA 25 datasets, showing no clear advantage.
> > >
> > > I strongly encourage the authors to revise Figure 4 in the final version, ensuring that all reported results are based on a fair evaluation protocol—specifically, without using the test set as validation. Please also provide corrected results for the UEA 25 datasets under this setting.
> > >
> > > Finally, while I do not have sufficient time to verify the results using the provided code, I hope the authors will make the final code publicly available to facilitate reproducibility and further research in this area. I believe that if the reported results can be reliably reproduced, this work will make a valuable contribution to multivariate time series classification.
> > >
> > > In light of these considerations, I have decided to raise my score to *Borderline Accept*.

---

> ### Author Response · Authors · 2025-08-04
>
> We sincerely thank for the valuable comments and suggestions, which helped us improve the quality of the paper. We are also grateful for the positive evaluation and the score adjustment. Your thoughtful feedback is greatly appreciated and will be reflected in the final version through updated experimental details and open-sourced results.
>
> Regarding **ShapeFormer**, while it uses 80% of the training data and reserves 20% as a validation set during the **shapelet discovery phase**, it **still uses the test set as the validation set during the final training stage**. This setup is explicitly described in the original paper’s **Experimental Implementation Details**.
>
> Our model was trained using the RAdam optimiser with an initial learning rate set as 0.01, a momentum of 0.9, and a weight decay of 5e-4. The training process involved a batch size of 16 for a total of 200 epochs. We configured the number of attention heads to be 16 and followed the protocol outlined in [47, 50]. **This protocol involves splitting the training set into 80% for training and 20% for validation, allowing us to fine-tune hyperparameters**. Once the hyperparameters were finalised, we conducted **model training on the entire training set and subsequently evaluated its performance on the designated official test set**.
>
> Moreover, the **public GitHub repository** of ShapeFormer confirms this usage: in `main.py`, **line 140** loads the `test_loader`, which is subsequently passed as the validation set on **lines 164 and 168**.
>
> Therefore, ShapeFormer follows the **same test-as-validation strategy** as other baselines. Under this setting, **SGN achieves 76.81% accuracy**, compared to **ShapeFormer’s 73.45%**, demonstrating SGN’s superior performance.
>
> We will update the experimental setup and related descriptions in the final version, and **release all results and code for full reproducibility**. We sincerely thank you again for your valuable comments and feedback.

---

### Official Review · Reviewer_Tx33 · 2025-07-02

**Clarity:** 2
**Significance:** 3
**Originality:** 3
**Rating:** 4
**Confidence:** 3

**Summary:**

This paper proposes SwinGroupNet (SGN), a novel approach for multivariate time series (MTS) classification, with key innovations including: Variable Group Embedding (VGE): A dynamic grouping strategy based on Brownian Distance Covariance (BDC) that partitions variables into clusters, enabling separate modeling of intra-group and inter-group dependencies to balance the trade-offs between independent and mixed variable modeling paradigms. Integrated periodic analysis and multi-scale convolution for simultaneous feature extraction across both temporal and variable dimensions. Periodic window shifting to enhance cross-window interactions and mitigate limitations of localized modeling.

**Questions:**

The proposed PWSM module exhibits strong dependence on periodic assumptions through its FFT-based windowing mechanism. This architectural choice may degrade model performance when processing non-periodic signals (e.g., white noise), (b) transient events (e.g., fault spikes in industrial data), or (c) irregular sampling scenarios - all common in real-world time series applications. What considerations did the author have regarding this?

**Ethical Concerns:**

["NO or VERY MINOR ethics concerns only"]

**Final Justification:**

Thanks for the responses.. I'll maintain my original rating.

**Limitations:**

1.Regarding the selection of baseline methods, we recommend incorporating state-of-the-art Multivariate Time Series representation approaches.
2. Evaluating the method's generalizability through additional experiments on fundamental time-series tasks such as forecasting and anomaly detection.

**Quality:**

3

**Strengths And Weaknesses:**

Strengths:
1. It demonstrates relatively excellent performance in multivariate time series classification tasks.

Weaknesses:
1. The architecture presented in Figure 2 is difficult to interpret. especially for the subfigures.
2. Line 142.  the final loss includes the L_task, in fact, the experimental validation appears limited to classification tasks.

---

> ### Author Rebuttal · Authors · 2025-07-31
>
> Thank you for your positive feedback. We're glad that you find our method demonstrates relatively excellent performance in multivariate time series classification tasks. We would like to provide some clarifications in the hope of addressing your concerns.
>
>
>
> ---
>
>
> **W1. The architecture presented in Figure 2 is difficult to interpret. especially for the subfigures.**
>
>
> **A1**. We apologize for the lack of clarity in the original model illustration, and we thank the reviewer for bringing this to our attention. Below, we provide a more detailed explanation of the SGN model architecture, which consists of **three main modules**: **VGE**, **MSWM**, and **PSPM**.
>
> - **VGE (Variable Grouping and Embedding)**:
>     - This module serves two purposes. First, it computes an **assignment matrix** based on intrinsic variable similarity, which is used to **fuse variables into groups**. These groups are then encoded using **group embeddings** to obtain compact representations.
>     - Second, the module applies **FFT** to estimate dominant periods and uses them to segment the time series into **periodic windows**.
>
> - **MSWM (Multi-Scale Window Module)**:
>     This module performs joint feature extraction along both the **temporal** and **variable** dimensions.
>
>     - First, in the temporal dimension, it applies **multi-scale depthwise convolutions** using multiple kernel sizes.
>
>     - Next, in the variable dimension, it performs **group-wise intra/inter pointwise convolutions** to capture both within-group and across-group interactions.
>
> - **PSPM (Period Shifting and Merging Module)**:
>     This module includes two components:
>
>     - **Period Merging**, which fuses features across adjacent windows to enhance temporal continuity.
>
>     - **Period Shifting**, which introduces dynamic offsets to the window boundaries, helping the model adapt to imperfect or misaligned periodicity.
>
>
> We hope this explanation provides a clearer understanding of our model architecture and its components.
>
>
> ---
>
>
>
> **W2. Line 142. the final loss includes the $\mathcal{L}_{\text{task}}$, in fact, the experimental validation appears limited to classification tasks.**
>
>
> **A2**. We apologize for the confusion. To clarify, the SGN model is primarily designed for **multivariate time series classification**.
>
> - $\mathcal{L}_{\text{task}}$​ corresponds to the **classification loss**, which serves as the main supervision signal. We will correct the symbols in subsequent versions.
>
> - $\mathcal{L}_{\text{sim}}$​ is a **similarity-based regularization term**, which encourages more structured and meaningful variable grouping by penalizing inconsistent similarity patterns.
>
>
>
> ---
>
>
>
> **Q1. The proposed PWSM module exhibits strong dependence on periodic assumptions through its FFT-based windowing mechanism. This architectural choice may degrade model performance when processing non-periodic signals (e.g., white noise), (b) transient events (e.g., fault spikes in industrial data), or (c) irregular sampling scenarios - all common in real-world time series applications. What considerations did the author have regarding this?.**
>
> **A3**. Thanks for your valuable insight. We would like to respond from the following two perspectives:
>
> 1. In the **UEA datasets used in Table 7 of Appendix C.3**, there are several datasets with **few variables** and **weak periodicity**, such as _EthanolConcentration_ and _Handwriting_, each containing only three variables. Despite these challenges, our method consistently outperforms baseline models, demonstrating its robustness in low-dimensional and weakly periodic settings.
> 2. While SGN is designed to leverage periodic patterns, it does **not rely rigidly on periodicity**. Instead, we incorporate a **dynamic shifting mechanism** that enables **local context fusion across windows**, allowing the model to remain effective even when the underlying periodic structure is weak or inconsistent.
>
> In addition, we have included more **multivariate datasets** and conducted comparisons against a broader range of **state-of-the-art multivariate methods** to further validate the effectiveness of our approach. The specific results are shown in the following table. **Wins/Draws/Losses** indicate the number of datasets (out of 30) on which the SGN method achieves higher, equal, or lower accuracy compared to the corresponding baseline methods. As shown, SGN consistently achieves **a greater number of first-place results and better average ranking** compared to state-of-the-art methods, demonstrating its robustness and generalization across diverse datasets.
>
>
> | Data/Model                | W.MUSE   | M.FCN |  TapNet  | ShapeNet | TodyNet  |   SVPT   | Shapeformer | MPTSNet  |   SGN(Ours)    |
> | ------------------------- | -------- | ----- | :------: | :------: | :------: | :------: | :---------: | :------: | :------: |
> | ArticularyWordRecognition | 99       | 97.3  |   98.7   |   98.7   |   98.7   | **99.3** |     99      |   97.7   |    99    |
> | AtrialFibrillation        | 33.3     | 26.7  |   33.3   |    40    |   46.7   |    40    |    53.3     |   53.3   | **66.7** |
> | BasicMotions              | **100**  | 95    | **100**  | **100**  | **100**  | **100**  |   **100**   | **100**  | **100**  |
> | CharacterTrajectories     | 99       | 98.5  | **99.7** |    98    |   N/A    |    99    |    99.2     |   N/A    |   99.6   |
> | Cricket                   | **100**  | 91.7  |   95.8   |   98.6   | **100**  | **100**  |    94.4     |   94.4   | **100**  |
> | DuckDuckGeese             | 57.5     | 67.5  |   57.5   | **72.5** |    58    |    70    |     64      |    68    |    64    |
> | EigenWorms                | 89       | 50.4  |   48.9   |   87.8   |    84    | **92.5** |     N/A     |   N/A    |   85.5   |
> | Epilepsy                  | **100**  | 76.1  |   97.1   |   98.7   |   97.1   |   98.6   |    98.6     |   97.1   |   97.8   |
> | EthanolConcentration      | 13.3     | 37.3  |   32.3   |   31.2   |    35    |   33.1   |    41.1     |   43.3   | **44.5** |
> | ERing                     | 43       | 13.3  |   13.3   |   13.3   |   91.5   |   93.7   |    87.4     |   94.4   | **95.9** |
> | FaceDetection             | 54.5     | 54.5  |   55.6   |   60.2   |   62.7   |   51.2   |    65.8     |   69.8   | **70.3** |
> | FingerMovements           | 49       | 58    |    53    |   58.9   | **67.6** |    60    |     55      |    64    |    64    |
> | HandMovementDirection     | 36.5     | 36.5  |   37.8   |   33.8   |   64.9   |   39.2   |    41.9     |   63.5   | **75.7** |
> | Handwriting               | **60.5** | 28.6  |   35.7   |   45.1   |   43.6   |   43.3   |    30.2     |   34.4   |   50.4   |
> | Heartbeat                 | 72.7     | 66.3  |   75.1   |   75.6   |   75.6   |    79    |  **81.5**   |   75.6   |   77.1   |
> | InsectWingbeat            | N/A      | 16.7  |   20.8   |    25    |   N/A    |   18.4   |    31.4     |   N/A    |  **68**  |
> | JapaneseVowels            | 97.3     | 97.6  |   96.5   |   98.4   |   N/A    |   97.8   |  **99.2**   |   98.6   |   98.9   |
> | Libras                    | 87.8     | 85.6  |    85    |   85.6   |    85    |   88.3   |  **95.5**   |   87.2   |   83.9   |
> | LSST                      | 59       | 37.3  |   56.8   |    59    |   61.5   | **66.6** |    63.8     |   60.4   |   63.7   |
> | MotorImagery              | 50       | 51    |    59    |    61    |    64    |  **65**  |     N/A     |  **65**  |  **65**  |
> | NATOPS                    | 87       | 88.9  |   93.9   |   88.3   |   97.2   |   90.6   |    96.1     |   94.4   | **98.3** |
> | PenDigits                 | 94.8     | 97.8  |    98    |   97.7   |   98.7   |   98.3   |  **99.1**   |   98.9   | **99.1** |
> | PEMS-SF                   | N/A      | 69.9  |   75.1   |   75.1   |    78    |   86.7   |     N/A     | **94.2** |   88.4   |
> | PhonemeSpectra            | 19       | 11    |   17.5   |   29.8   | **30.9** |   17.6   |    29.3     |   14.4   |   23.1   |
> | RacketSports              | **93.4** | 80.3  |   86.8   |   88.2   |   80.3   |   84.2   |    88.8     |   87.5   | **93.4** |
> | SelfRegulationSCP1        | 71       | 87.4  |   65.2   |   78.2   |   89.8   |   88.4   |    91.8     |   92.8   | **93.9** |
> | SelfRegulationSCP2        | 46       | 47.2  |    55    |   57.8   |    55    |    60    |    56.1     |   57.2   | **60.6** |
> | SpokenArabicDigits        | 98.2     | 99    |   98.3   |   97.5   |   N/A    |   98.6   |  **99.7**   |   99.5   | **99.7** |
> | StandWalkJump             | 33.3     | 6.7   |    40    |   53.3   |   46.7   |   46.7   |  **66.7**   |   53.3   |   53.3   |
> | UWaveGestureLibrary       | 91.6     | 89.1  |   89.4   |   90.6   |    85    | **94.1** |     90      |   88.1   |   92.2   |
> | Average rank              | 5.57     | 7.17  |   6.13   |   4.80   |   4.27  |   3.70   |    3.19     |   3.74   | **2.13** |
> | Number of top-1           | 5        | 0     |    2     |    2     |    4     |    7     |      7      |    3     |  **15**  |
> | Wins                      | 20       | 28    |    27    |    23    |    21    |    19    |     15      |    20    |    -     |
> | Draws                     | 4        | 0     |    1     |    2     |    2     |    3     |      5      |    4     |    -     |
> | Loses                     | 4        | 2     |    2     |    5     |    3     |    8     |      7      |    3     |    -     |

---

> > ### Comment · Reviewer_Tx33 · 2025-08-04
> >
> > The authors addressed my concerns.

---

> > > ### Author Response · Authors · 2025-08-04
> > >
> > > Thank you for your careful review of our work. We are glad to have addressed your concerns and sincerely appreciate your recognition and constructive feedback. We will incorporate the corresponding updates in the revised version of the paper.

---

### Official Review · Reviewer_86Jq · 2025-07-08

**Clarity:** 3
**Significance:** 3
**Originality:** 2
**Rating:** 5
**Confidence:** 3

**Summary:**

The paper introduces SwinGroupNet (SGN), a novel framework for multivariate time series (MTS) classification that explicitly models both intra- and inter-variable dependencies using a grouped embedding approach and hierarchical temporal feature extraction. By combining three components—Variable Group Embedding (VGE), Multi-Scale Group Window Mixing (MGWM), and Periodic Window Shifting and Merging (PWSM)—SGN aims to preserve semantic structures, address heterogeneous variable types, and exploit periodic temporal patterns. Experimental results on multiple benchmark datasets show promising improvements over prior work.

**Questions:**

1- How does the computational cost (in terms of FLOPs, training time, inference time, and memory usage) of SGN compare to the baseline models?
2- Does the model generalize well to datasets with fewer variables or less clear periodic structure?
The reliance on periodic patterns in PWSM might not apply broadly.

**Ethical Concerns:**

["NO or VERY MINOR ethics concerns only"]

**Final Justification:**

The authors have clarified more detailed about implementation and baseline comparison

**Quality:**

2

**Strengths And Weaknesses:**

# Strengths

1. **Clear Motivation**: The paper convincingly discusses the limitations of univariate decomposition and full multivariate modeling approaches in MTS.
2. **Novel Architecture**: The use of group-based embeddings and hierarchical temporal modeling (through PWSM and MGWM) is an interesting and original idea.
3. **Strong empirical results**: Demonstrates consistent state-of-the-art performance across diverse benchmarks with a reported 4.2% improvement on average.
4. **Code Availability**: Open-sourcing the implementation increases reproducibility and community impact.

# Weaknesses
- **Architectural Complexity Without Computational Justification**

    While SGN demonstrates strong performance, the proposed design—incorporating Variable Group Embedding (VGE), Multi-Scale Group Window Mixing (MGWM), and Periodic Window Shifting and Merging (PWSM)—introduces considerable architectural complexity. However, the paper **does not provide any analysis of computational cost** (e.g., training/inference time, parameter count, FLOPs, or GPU memory usage), which is critical when evaluating practical deployment, especially for long multivariate sequences. Without this, it is difficult to assess the **efficiency vs. performance tradeoff** compared to simpler or more lightweight baselines.

- **Lack of Controlled Comparison With Existing Channel Dependency Methods**

    Although the motivation for capturing inter-variable dependency is valid, the paper does not thoroughly compare SGN with **alternative methods for modeling channel correlations**. For instance, prior work like MSGNet (AAAI 2024) has explored efficient multi-scale inter-series correlation modeling. Additionally, **no study is conducted using a common architectural backbone** to test whether simpler methods (e.g., channel attention, global attention, temporal convolutions) underperform consistently compared to SGN.

- **Scalability**: No discussion is included on the computational complexity, especially for large MTS with many variables and long sequences.

---

> ### Author Rebuttal · Authors · 2025-07-31
>
> Many thanks for your valuable and encouraging feedback. We sincerely appreciate that you found our paper to be well-motivated, with a novel architecture and strong empirical performance. We would like to provide further clarifications to address your comments in more detail.
>
> ---
>
> **W1. Architectural Complexity Without Computational Justification**
>
> **A1**. We appreciate the reviewer’s concern regarding computational efficiency. For the input time series data $X \in \mathbb{R}^{d_{\text{m}} \times \text{group} \times L}$, where $d_{\text{m}}$​ denotes the embedding dimension and *L* is the temporal length (with $L>>d_{\text{m}}$​​). We adopt a combination of **depthwise and pointwise convolutions** for feature extraction. This design allows for efficient modeling with the following complexity: **Time complexity** $\mathcal{O}(L \cdot d_{\text{m}}^2)$ and **Space complexity** $\mathcal{O}(d_{\text{m}}^2)$. A comparison with other methods is detailed in the table below:
>
> | Methods     | Time Complexity                                  | Space complexity  |
> | ----------- | ------------------------------------------------ | ----------------- |
> | SGN（Ours）   | $O(Ld_m)$                                        | $O(Ld_m)$         |
> | TVNet       | $O(Ld_m^2)$                                      | $O(d_m^2 + Ld_m)$ |
> | MICN        | $O(Ld_m^2)$                                      | $O(Ld_m^2)$       |
> | FEDformer   | $O(L)$                                           | $O(L)$            |
> | Crossformer | $O\left(\frac{d_m}{L_{\text{seg}}^2} L^2\right)$ | $O(Ld_m)$         |
>
>
> We have also conducted a detailed comparison with baseline methods in terms of **FLOPs** and **GPU memory usage**, as shown in **Appendix E**. The results confirm that our approach achieves favorable computational efficiency while maintaining strong performance.
>
>
> ---
>
> **W2. Lack of Controlled Comparison With Existing Channel Dependency Methods.**
>
>
> **A2**. We sincerely appreciate your insightful suggestion. Following your advice, we have included additional baseline methods for a more comprehensive comparison. Specifically, we categorized these methods into the following groups:
>
> - **Multi-scale methods**: MSGNet, MedFormer
> - **Channel attention methods**: iTransformer, Crossformer
> - **Global attention methods**: Transformer, PatchTST
> - **Temporal convolution methods**: TCN, ModernTCN
>
> This categorization allows for a more structured evaluation, and the extended comparisons further demonstrate the effectiveness and robustness of our proposed method. The specific results are shown in the following table:
>
> | Dataset       | TDBRAIN  |          | **FLAAP** |           | **UCI-HAR** |           | **PTB-XL** |           |
> | ------------- | :------: | :------: | :-------: | :-------: | :---------: | :-------: | :--------: | :-------: |
> | Metric        | Accuracy |    F1    | Accuracy  |    F1     |  Accuracy   |    F1     |  Accuracy  |    F1     |
> | MSGNet        |  85.64   |  85.63   |   74.38   |   74.01   |    92.43    |   92.84   |   69.96    |   57.64   |
> | Medformer     |  89.62   |  89.62   |    74     |   73.84   |    90.17    |   90.27   |   72.87    |   62.02   |
> | Crossformer   |  81.56   |   81.5   |   76.33   |   76.14   |    90.66    |   90.68   |    73.3    |   62.59   |
> | iTransformer  |  74.67   |  74.65   |   75.83   |   75.57   |    93.47    |   93.46   |   69.28    |   56.2    |
> | TCN           |   88.5   |  88.42   |   76.11   |   75.79   |    92.72    |   92.66   |   72.67    |   62.04   |
> | MordenTCN     |   87.6   |  87.54   |   71.66   |   71.37   |    92.75    |   92.8    |   72.85    |   61.33   |
> | Transformer   |  87.17   |   87.1   |   74.14   |   73.71   |    90.68    |   90.69   |   70.59    |   59.05   |
> | PatchTST      |  79.25   |   79.2   |   56.23   |   55.57   |    86.83    |   87.17   |   73.23    |   62.61   |
> | **SGN(ours)** | **99.9** | **99.9** | **80.81** | **80.35** |  **95.62**  | **95.64** |  **73.8**  | **63.43** |
>
>
> ---
>
>
> **W3. No discussion is included on the computational complexity, especially for large MTS with many variables and long sequences.**
>
>
> **A3**. We sincerely thank the reviewer for the thoughtful suggestion. Our SGN model effectively reduces computational complexity by combining **periodic windowing and fusion mechanisms** for long sequences, along with the use of **depthwise and pointwise convolutions**.
> As shown in **Table 7 of Appendix C.3**, our method outperforms other baselines even on challenging datasets such as EthanolConcentration(with a sequence length of 1751) and FaceDetection(with 144 variables), demonstrating strong performance on both long sequences and high-dimensional multivariate data.
>
>
> ---
>
> **Q1. How does the computational cost (in terms of FLOPs, training time, inference time, and memory usage) of SGN compare to the baseline models**
>
> **A4.** We have provided the corresponding response in **A1 of W1**.
>
> ---
>
> **Q2. Does the model generalize well to datasets with fewer variables or less clear periodic structure? The reliance on periodic patterns in PWSM might not apply broadly.**
>
>
> **A5**. Thanks for your valuable insight. We would like to respond from the following two perspectives:
>
> 1. In the **UEA datasets used in Table 7 of Appendix C.3**, there are several datasets with **few variables** and **weak periodicity**, such as _EthanolConcentration_ and _Handwriting_, each containing only three variables. Despite these challenges, our method consistently outperforms baseline models, demonstrating its robustness in low-dimensional and weakly periodic settings.
> 2. While SGN is designed to leverage periodic patterns, it does **not rely rigidly on periodicity**. Instead, we incorporate a **dynamic shifting mechanism** that enables **local context fusion across windows**, allowing the model to remain effective even when the underlying periodic structure is weak or inconsistent.

---

> > ### Comment · Reviewer_86Jq · 2025-08-09
> > **Response to Authors**
> >
> > Thanks to the authors for the detailed response. While addressed part of my concerns, my issue of unified backbone is yet to be addressed, I will keep my scores

---

> > > ### Author Response · Authors · 2025-08-09
> > >
> > > We sincerely thank the reviewer for the valuable comments and constructive feedback. We would like to clarify that we implemented our method and all the baseline models within the unified framework provided by the **Time-Series-Library project** from Tsinghua University, following the same code structure to ensure fairness and consistency in experimental settings.

---

> ### Author Response · Authors · 2025-08-06
>
> Dear Reviewer,
>
> We would like to kindly follow up regarding our rebuttal. We value your feedback greatly and would appreciate it if you could share any additional comments or questions when convenient. We are happy to provide further clarifications, experiments, or supporting materials at any time to facilitate the discussion.
>
> Thank you very much for your time and consideration.

---

### Note · Authors · 2025-08-11

**Dear Reviewers and AC,**

We sincerely thank you for your time and constructive feedback, which have greatly helped us improve our work.

SGN is built on a convolutional architecture, with VGE for variable group & periodic window extraction, MGWM for intra-/inter-group interactions and multi-scale temporal feature learning, and PWSM for inter-window interactions that expand local features to global context, allowing the model to remain effective even when the underlying periodic structure is weak or inconsistent. These modules jointly balance high performance and efficiency.

All our main experiments are conducted within the **unified Time-Series-Library framework**, ensuring a fair comparison with baselines. On four benchmark datasets, SGN improves average accuracy by **4.2%** over SOTA methods (transformer-, CNN-, and MLP-based). On UEA benchmarks, **under the same experimental setup**:

- **UEA-10:** SGN achieves **77.6%**, surpassing TimeMixer++ (75.3%) and MPTSNet (75.4%).

- **UEA-25:** SGN achieves **76.81%**, outperforming MPTSNet (74.00%) and Shapeformer (73.45%).


These results clearly demonstrate that SGN exhibits **strong performance and robust generalization** across diverse datasets. In addition, we conduct **comprehensive ablation studies** to rigorously validate the effectiveness of our architecture and each individual module.

We will update the paper to clearly describe experimental settings, include both training-validation splits, and release full code/scripts for reproducibility.

**Once again, we truly appreciate your detailed reviews and valuable suggestions.**

---

### Decision · Program_Chairs · 2025-09-17

**Decision:**

Accept (poster)

**Comment:**

The paper proposes a method for multivariate timeseries classification which works by grouping features, modeling inter and intra-group dependencies and learning multi-scale features. Most reviewers praised the paper's novelty, strong empirical results and clarity of presentation.

The comparison against other channel dependency methods is appreciated, as are the additional experimental results presented in response to reviewers eNCr and 6yyx.

During the response period, the authors addressed the reviewer's concerns, leading to a unanimous decision to accept the paper.